# KRIS-Bench: Benchmarking Next-Level Intelligent Image Editing Models

**Yongliang Wu**[1,4*]   **Zonghui Li**[1]   **Xinting Hu**[2†]   **Xinyu Ye**[3]   **Xianfang Zeng**[4‡]
**Gang Yu**[4]   **Wenbo Zhu**[5]   **Bernt Schiele**[2]   **Ming-Hsuan Yang**[6]   **Xu Yang**[1]

[1] Southeast University   [2]Max Planck Institute for Informatics   [3] Shanghai Jiao Tong University
[4] StepFun   [5] University of California, Berkeley   [6] University of California, Merced
Project Page: https://yongliang-wu.github.io/kris_bench_project_page/

## Abstract

Recent advances in multi-modal generative models have enabled significant progress in instruction-based image editing. However, while these models produce visually plausible outputs, their capacity for knowledge-based reasoning editing tasks remains under-explored. In this paper, We introduce **KRIS-Bench** (**K**nowledge-based **R**easoning in **I**mage-editing **S**ystems **Bench**mark), a diagnostic benchmark designed to assess models through a cognitively informed lens. Drawing from educational theory, KRIS-Bench categorizes editing tasks across three foundational knowledge types: *Factual*, *Conceptual*, and *Procedural*. Based on this taxonomy, we design 22 representative tasks spanning 7 reasoning dimensions and release 1,267 high-quality annotated editing instances. To support fine-grained evaluation, we propose a comprehensive protocol that incorporates a novel *Knowledge Plausibility* metric, enhanced by knowledge hints and calibrated through human studies. Empirical results on 10 state-of-the-art models reveal significant gaps in reasoning performance, highlighting the need for knowledge-centric benchmarks to advance the development of intelligent image editing systems.

## 1   Introduction

Recent advances in multi-modal generative models have led to impressive performance in instruction-based image editing [1–3]. Given various textual prompts, these models can produce visually coherent and semantically aligned edits across tasks such as object manipulation [4, 5], style transformation [6, 7], and action simulation [8, 9]. However, while the editing quality of these model outputs has improved substantially, the reasoning processes underpinning such edits remain under-explored [10–13]. For example, as shown in Figure 1 (b), when given the instruction *"add a piece of solid sodium to the water"*, the models generate a visually plausible image in which the sodium appears submerged in the water. But it reveals a lack of reasoning grounded in chemistry knowledge, as solid sodium will react violently with water, releasing a large amount of heat that causes the water to boil. Successful reasoning may require perceptual recognition, spatial interpretation, social commonsense, science concepts, or procedural planning [14, 15]. The diversity of these knowledge types underscores the need for more fine-grained and cognitively informed evaluation frameworks that can systematically disentangle the reasoning capabilities required for different editing goals [16–18].

Recently, several benchmarks have been proposed to evaluate the capabilities of image editing models [3, 4, 8, 19–26]. *RISEBench* [26], most relevant to our work, introduces reasoning-aware image

---

*Work done during an internship at StepFun

†Corresponding author

‡Project leader

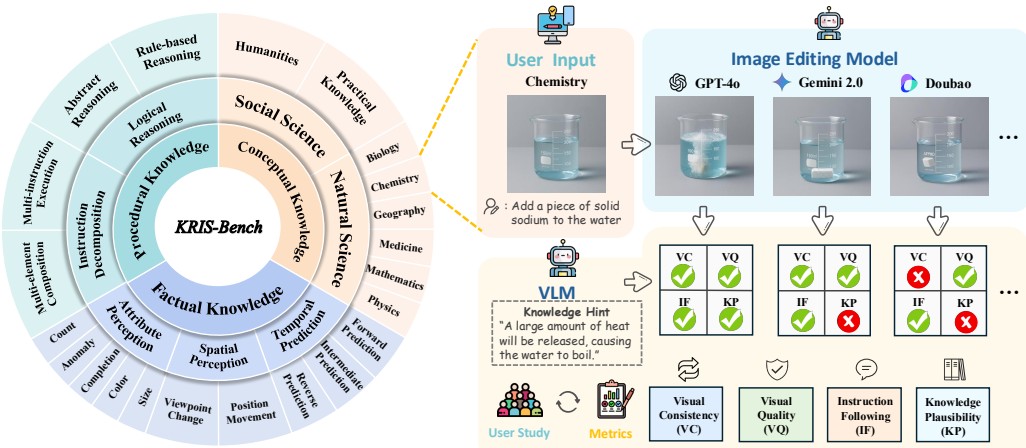

(a) The taxonomy of KRIS-Bench.    (b) The evaluation process and metrics of KRIS-Bench.

Figure 1: (a) We present KRIS-Bench, a benchmark for instruction-based image editing grounded in a knowledge-based reasoning taxonomy. It covers 3 knowledge dimensions, 7 reasoning dimensions, and 22 editing tasks. Specific examples are shown in Figure 2. (b) Given an editing pair of (image, instruction) under a specific reasoning dimension (*i.e.*, Chemistry in Natural Science), we evaluate the output of image editing models with automated VLM tools over the proposed four complementary metrics, which are aligned with human scoring.

editing evaluations across temporal, causal, spatial, and logical dimensions. However, its reasoning types remain coarse and do not provide a formal structure for representing the underlying knowledge required by different tasks. Rather than simply evaluating image editing through task categories or action types, we benchmark it based on a structured understanding of knowledge [14]. We view instruction-based image editing as a cognitively grounded process that mirrors human learning. From this perspective, equipping image editing models with the ability to identify, internalize, and apply appropriate knowledge during editing resembles the process of educating a student to perceive, reason about, and interact with the real world. Guided by this analogy, we draw inspiration from the revised taxonomy of educational objectives proposed by Anderson and Krathwohl [27], and define three foundational types of knowledge: *Factual knowledge*, *Conceptual knowledge*, and *Procedural knowledge*. This taxonomy supports a systematic decomposition of the knowledge demands involved in the reasoning process of image editing, and provides a principled foundation for our design of diagnostic benchmarks for image editing models [28].

Building on these knowledge types, we present **KRIS-Bench** (**K**nowledge-based **R**easoning in **I**mage-editing **S**ystems **Bench**mark), a diagnostic benchmark designed to systematically evaluate the reasoning capabilities of image editing models. KRIS-Bench adopts a top-down design paradigm grounded in principles of cognitive education. It structures tasks according to three foundational knowledge types, each further decomposed into specific reasoning dimensions. For example, factual knowledge covers directly observable properties and does not involve abstract inference or contextual interpretation, thus supporting basic reasoning processes such as perceptual recognition [7], spatial relation understanding [29], and temporal prediction [30]. The taxonomy is visualized in Figure 1 (a), where 22 editing tasks are organized across 7 reasoning dimensions under the three knowledge types. To support reliable evaluation at scale, KRIS-Bench comprises 1,267 high-quality instances.

Furthermore, we propose a comprehensive evaluation protocol grounded in vision-language models (VLMs) [31–35]. Beyond conventional metrics [26, 36–38], we introduce a new dimension, *Knowledge Plausibility*, which assesses whether the edited outputs align with real-world knowledge, as illustrated in Figure1 (b). To facilitate this evaluation, each knowledge-intensive test case is accompanied by a manually curated *knowledge hint* designed to guide the VLM's reasoning. We conduct a user study to validate the alignment between our evaluation protocol and human judgments, and demonstrate that the inclusion of knowledge hints significantly enhances the plausibility assessment by VLMs [39–41]. Extensive experiments across 10 state-of-the-art models reveal persistent limitations in performing knowledge-grounded reasoning for image editing tasks.

The main contributions of this work are:

- We propose the first cognitively grounded taxonomy of knowledge types for instruction-based image editing. Drawing from educational theory, we systematically define *Factual*, *Conceptual*, and *Procedural* knowledge as the foundation for evaluating reasoning capabilities.
- We introduce **KRIS-Bench**, a comprehensive benchmark consisting of 22 carefully designed tasks across 7 reasoning dimensions, supported by 1,267 expertly annotated editing instances. This significantly expands the scale and depth of reasoning evaluation in the image editing.
- We design a comprehensive evaluation protocol that, for the first time, introduces the *Knowledge Plausibility* dimension to assess whether model-generated edits are consistent with real-world knowledge, with manually curated *knowledge hints* to support more reliable plausibility judgments.
- We conduct systematic experiments of 10 state-of-the-art image editing models, revealing substantial limitations across knowledge types, reasoning dimensions, and editing tasks.

## 2  Related Work

**Instruction-based Image Editing Methods.**  Instruction-based image editing [17, 42, 43] has progressed significantly through the use of diffusion models and instruction-following strategies. Some methods enable test-time controllability by altering the diffusion trajectory, including partial denoising from intermediate steps [44], attention-based control for localized edits [45], CLIP-guided manipulation with region-of-interest masks [46, 47], and latent inversion strategies that optimize noise embeddings to preserve fidelity [48].  Beyond test-time control, many approaches improve editing performance through model training or fine-tuning. Some enhance the architecture with task-aware conditioning, cross-modal attention, or instruction-parsing modules to support more complex edits [25, 49, 50].  Others scale up with large-scale instruction tuning on millions of image-text pairs to boost generalization and fidelity for open-ended prompts [4, 51].  A further line of work incorporates human feedback via reward learning or reference-based alignment to better capture user intent [52, 53].  Closed-source systems such as GPT-Image-1 [54], Doubao [55], and Gemini 2.0 Flash Experimental [56] further push performance through large-scale multi-modal training and integrated reasoning. However, across both open and closed models, existing methods emphasize visual plausibility and instruction adherence, with limited attention to the knowledge and reasoning processes essential for cognitively grounded editing.

**Benchmarks for Instruction-based Image Editing.**  To effectively evaluate the capabilities of instruction-based image editing models [57–62], a growing number of datasets and benchmarks have been proposed. *EditBech* [19], *TEdBench* [21], and *EditEval* [3] focus on task-oriented evaluation, targeting canonical sub-tasks such as inpainting, attribute manipulation, or layout adjustment. To expand evaluation coverage, benchmarks like *EMU-Edit*[25], *GEdit-Bench* [8], and *REALEDIT* [38] collect diverse free-form user instructions, while *I2EBench* [22] scales across editing types and metrics. *Complex-Edit* [63] further introduces multi-step editing chains to model task complexity. Despite these advances, these work focus on task complexity or data scale, without explicitly modeling the reasoning processes or knowledge structures involved in instruction understanding. Recent works try to address this gap by incorporating reasoning-aware evaluation [24]. *AURORA-BENCH* [64] focuses on action-centric edits by leveraging curated triplets from videos and simulations, and *SmartEdit* [13] explores spatial and interaction-based reasoning within ambiguous editing scenarios. IntelligentBench [65] is designed to evaluate the ability of editing models in complex multimodal reasoning, but it does not provide a detailed categorization of task types. *RISEBench* [26] categorizes tasks along temporal, causal, spatial, and logical dimensions. However, these reasoning axes remain coarse and are not grounded in a formal cognitive or knowledge-based framework, limiting their capacity to capture the full scope of reasoning challenges in instruction-driven image editing.

## 3  KRIS-Bench

In this section, we introduce **KRIS-Bench**, a comprehensive benchmark designed to evaluate image editing models through the lens of knowledge-based reasoning. A comparative analysis with prior reasoning-based image editing benchmarks is presented in Table 1. **KRIS-Bench** offers the most comprehensive coverage to date, featuring the largest size (1,276 samples across 22 tasks) with a strong emphasis on reasoning capabilities across varying levels of complexity. For cases involving knowledge-based reasoning, we additionally provide knowledge hints to assist the evaluation process.

Table 1: Comparison of open-source reasoning-based image editing benchmarks.

| Dataset | Publication | Size | Dimensions | Tasks | Complexity | Knowledge Hints |
|---|---|---|---|---|---|---|
| AURORA-Bench [64] | NeurIPS 2024 | 400 | – | 8 | Simple | ✗ |
| SmartEdit [13] | CVPR 2024 | 219 | 2 | 7 | Medium | ✗ |
| RISE [26] | NeurIPS 2025 | 360 | 4 | 16 | Hard | ✗ |
| IntelligentBench [65] | arXiv 2025.5 | 350 | – | – | Medium | ✗ |
| **KRIS-Bench** | – | **1,267** | **7** | **22** | Mixed | ✓ |

## 3.1 Taxonomy of Knowledge Types

Our knowledge-based reasoning taxonomy in image editing models is inspired by the revised Bloom taxonomy of educational objectives [27]. We organize the knowledge required in image editing into three levels: *Factual Knowledge*, *Conceptual Knowledge*, and *Procedural Knowledge*.[4] Unlike prior works that emphasize editing actions, our focus is on the types of knowledge a model must internally represent and apply to perform a reasoning-aware edit. This perspective is rooted in pedagogical theory, where different levels of knowledge serve as a foundation for learning and problem solving.

**Factual Knowledge** includes directly observable properties such as visual attributes (*e.g.*, color, size), spatial relations (*e.g.*, left/right, different viewpoint), and temporal cues (*e.g.*, before/after states). This knowledge does not require abstract inference or contextual interpretation, serving as the basic prerequisite for more complex reasoning.

**Conceptual Knowledge** represents a higher-order form of understanding that connects perceptual information to generalizable principles from the physical, biological, or social world. Unlike factual recognition, conceptual knowledge enables models to anticipate plausible outcomes following real-world dynamics, knowledge, and rules. For example, the instruction "Ripen the bananas by turning them yellow" presumes an understanding of the natural ripening process.

**Procedural Knowledge** refers to the ability of a model to perform multi-step reasoning, task decomposition, and rule-based execution within image editing contexts. It involves not only understanding what change should occur, but also how to perform that change in procedure. Procedural knowledge is essential for instructions requiring multi-element coordination (*e.g.*, multi-element referring generation) or complex logical reasoning (*e.g.*, complete the Raven's progressive matrix) [66].

## 3.2 Knowledge-Based Task Formulation

Drawing from the three knowledge types, we define 7 associated reasoning dimensions that correspondingly span across 22 tasks. The tasks in KRIS-Bench are not mere isolated editing actions. Instead, they are crafted and organized based on their specific knowledge requirements derived from our taxonomy. Representative examples from each task are illustrated in Figure 2.

**Factual Knowledge.** Tasks in this category evaluate fundamental visual and temporal understanding that does not require external knowledge or reasoning. The sub-dimensions encompass:

- **Attribute Perception.** Modifications to object count, color, size, part completion, and correction of abnormalities based on direct perception in the image.
- **Spatial Perception.** Movement of objects to target locations within the image and adjustment of viewpoints for the same object.
- **Temporal Prediction.** Prediction of previous, intermediate, or future frames based on surrounding frames for maintaining temporal consistency.

**Conceptual Knowledge.** Tasks in this category necessitate understanding and applying real-world knowledge beyond perceptual cues. The sub-dimensions encompass:

- **Social Science:** Modifications involving commonsense reasoning (*e.g.*, adjusting a clock for daylight saving time) and edits based on cultural or religious contexts (*e.g.*, substituting a dish with mooncakes for a festival).
- **Natural Science:** Modifications based on science principles, covering biology (*e.g.*, fruit ripening), chemistry (*e.g.*, color changes in pH indicators), geography (*e.g.*, terrain alterations), mathematics

---

[4]We do not include Metacognitive Knowledge in Bloom's taxonomy, as it involves self-monitoring and learning regulation, which current large models do not yet demonstrate within the one-turn image editing process.

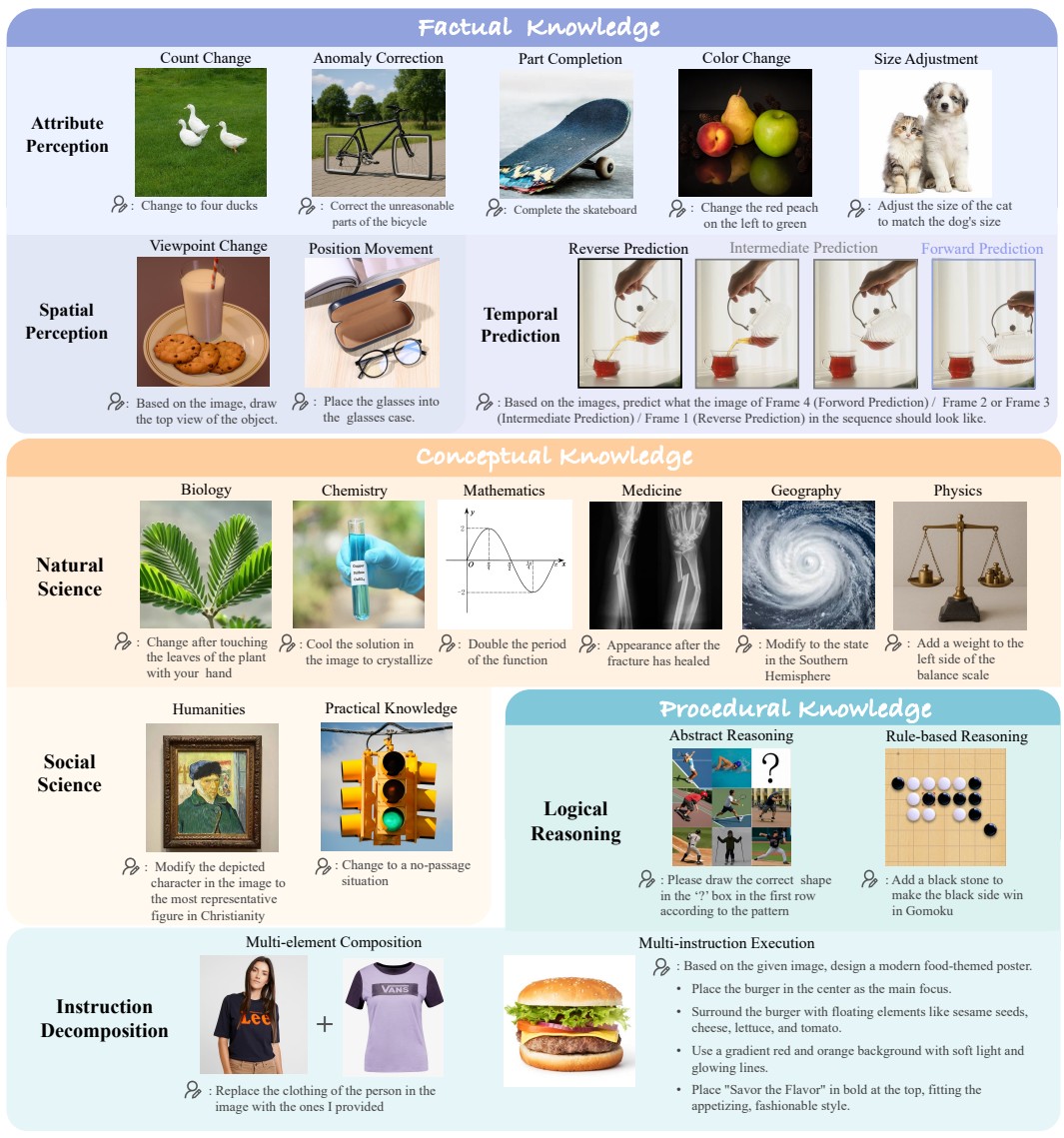

Figure 2: Representative examples from the 22 knowledge-based reasoning image editing tasks in KRIS-Bench. Each task is designed to evaluate specific knowledge grounded in factual, conceptual, or procedural, covering diverse reasoning dimensions.

(*e.g.*, geometric transformations), medicine (*e.g.*, blood pressure changes), and physics (*e.g.*, changes based on physical laws).

**Procedural Knowledge.** Tasks in this category involve executing structured reasoning processes and following multi-step instructions. The sub-dimensions include:

- **Logical Reasoning:** Modifications involving reasoning with symbolic structures and numerical relationships (*e.g.*, solving puzzles or applying logical rules).
- **Instruction Decomposition:** Modifications requiring the execution of multiple sequential instructions (*e.g.*, designing a poster) and integrating visual elements from various sources into a coherent scene (*e.g.*, combining objects from different images).

## 3.3 Data Collection

Most images in our benchmark were collected from the internet, with a small portion generated using generative models [54] and collected from existing datasets [13, 67–72]. For each image, one editing instruction is created by trained annotators. To enhance instruction diversity and realism, we augment

the original prompts using ChatGPT, paraphrasing and elaborating them under human supervision. The data was curated by three human annotators, two of whom have obtained Bachelor's degrees, while the third is currently pursuing one. All annotations were subsequently reviewed by three experts with Ph.D. degrees. For tasks requiring domain expertise (*e.g.*, physics-based or biomedical edits), additional domain-specific reviewers were consulted.

### 3.4 Evaluation Metrics

To comprehensively evaluate the performance of state-of-the-art image editing models on KRIS-Bench, we propose a four-dimensional evaluation protocol. In addition to the three widely adopted dimensions, namely *Visual Consistency*, *Visual Quality*, and *Instruction Following* [13, 24, 26, 73], we introduce a novel fourth dimension called *Knowledge Plausibility*, which explicitly assesses whether the generated edits are consistent with real-world knowledge. To support this evaluation, we provide a concise *knowledge hint* for test cases that require real-world knowledge. Each hint is a brief description of the expected outcome based on humanities, scientific, or procedural understanding. For example, *adding purple cabbage indicator to acidic water should result in a red color change*. These hints offer evaluators the necessary reference to determine whether the edited image reflects plausible and knowledge-consistent effects.

**Visual Consistency.** This dimension evaluates whether the edited image faithfully preserves the parts of the original image that are not semantically or spatially related to the instruction. An effective editing model should localize changes precisely while leaving the rest of the scene unchanged.

**Visual Quality.** This dimension evaluates the perceptual quality of the generated image, focusing on overall realism, natural appearance, and the absence of noticeable artifacts. It assesses whether the output maintains structural coherence and visual plausibility, without introducing distortions such as unnatural textures, broken geometry, or degraded fine details.

**Instruction Following.** This dimension evaluates whether the model accurately and completely executes the user-provided instruction. It focuses purely on the literal fulfillment of the editing instruction, independent of perceptual quality or real-world plausibility. For instance, when given the instruction "add a wooden block into the tank", this dimension solely verifies if the edited image includes the additional wooden block in the tank, without regard to whether the block floats or sinks.

**Knowledge Plausibility.** This dimension assesses whether the edits are consistent with real-world knowledge and domain-specific principles. It functions as a higher-level criterion that evaluates the coherence of the output within a plausible environment. For example, the addition of a wooden block to a tank that appears fully submerged indicates poor plausibility of the physics knowledge. Edits that fail to fulfill the instruction are automatically considered implausible under this dimension, as basic instruction compliance is a prerequisite for meaningful knowledge reasoning. This metric is only available for tasks in Natural Sciences, Social Sciences, and Logical Reasoning.

Each evaluation metric is rated from 1 to 5. We use GPT-4o (May 2025) as the evaluation model, with carefully crafted prompts tailored for each dimension to ensure precise and consistent assessment [74].

## 4 Experiments and Analysis

### 4.1 Evaluation Models & Settings

We evaluate 10 state-of-the-art image editing models on KRIS-Bench to assess their reasoning capabilities. These models include three closed-source models: GPT-Image-1 [54](GPT-4o), Gemini 2.0 Flash Experimental [56](Gemini 2.0), and Doubao [55].[5] Seven open-source models: OmniGen [75], Emu2 [76], BAGEL [65], Step1X-Edit [8], AnyEdit [9], InstructPix2Pix [5](InsPix2Pix), and MagicBrush [4]. Note that open-source models, except OmniGen and Emu2, are limited to single-image inputs and thus cannot be evaluated on tasks requiring multiple input images. In such tasks, these models receive an evaluation score of one (lowest). Moreover, BAGEL is capable of performing image editing in reasoning mode, which we refer to as BAGEL-Think in our experiments. All generation and evaluation processes were conducted on H100 GPUs, using default hyperparameter settings to ensure fairness and reproducibility.

---

[5]Results obtained via OpenAI, API Google AI Studio, and Doubao App (all in April 2025).

Table 2: Performance of different models across different reasoning dimensions and metrics, including Visual Consistency (VC), Visual Quality (VQ), Instruction Following (IF), and Knowledge Plausibility (KP). Scores marked with ∗ indicate models unable to handle multi-image input tasks, with the corresponding task scores set to 0. The performance of open-source and closed-source models is separately marked with the best performance in **bold**, and the second best underlined.

| Reasoning Dimension | Metric | Closed-Source Models | | | Open-Source Models | | | | | | | |
|---|---|---|---|---|---|---|---|---|---|---|---|---|
| | | GPT-4o | Gemini 2.0 | Doubao | OmniGen | Emu2 | BAGEL | BAGEL-Think | Step1X-Edit | AnyEdit | MagicBrush | InsPix2Pix |
| **Factual Knowledge** | | | | | | | | | | | | |
| Attribute Perception | VC | **74.50** | 69.50 | 66.75 | 35.75 | 47.75 | 66.75 | **74.75** | 63.00 | 54.75 | 53.50 | 17.50 |
| | VQ | **94.75** | 81.75 | 89.00 | 49.50 | 75.25 | 67.00 | 75.00 | 70.25 | 67.50 | **76.25** | 55.50 |
| | IF | **80.25** | 47.75 | 57.00 | 28.50 | 31.50 | 40.50 | **49.50** | 33.25 | 20.75 | 32.00 | 18.00 |
| | Avg | **83.17** | 66.33 | 70.92 | 37.92 | 51.50 | 58.08 | **66.42** | 55.50 | 47.67 | 53.92 | 30.33 |
| Spatial Perception | VC | **69.50** | 60.50 | 67.50 | 24.00 | 41.50 | 53.50 | **77.25** | 64.25 | 55.75 | 38.00 | 13.25 |
| | VQ | **94.50** | 83.25 | 89.00 | 50.00 | 77.75 | 71.25 | 81.25 | **83.00** | 72.00 | 69.25 | 40.25 |
| | IF | **73.25** | 46.25 | 21.00 | 10.75 | 18.25 | 38.75 | **44.75** | 8.00 | 7.75 | 11.50 | 10.50 |
| | Avg | **79.08** | 63.33 | 59.17 | 28.25 | 48.83 | 54.50 | **67.75** | 51.75 | 45.17 | 39.58 | 21.33 |
| Temporal Prediction | VC | 54.00 | **54.50** | 26.75 | **19.25** | 12.50 | 0.00* | 0.00* | 0.00* | 0.00* | 0.00* | 0.00* |
| | VQ | **86.25** | 75.00 | 77.50 | 26.25 | **37.50** | 0.00* | 0.00* | 0.00* | 0.00* | 0.00* | 0.00* |
| | IF | **64.50** | 62.25 | 17.50 | **20.00** | 16.50 | 0.00* | 0.00* | 0.00* | 0.00* | 0.00* | 0.00* |
| | Avg | **68.25** | 63.92 | 40.58 | 21.83 | **22.17** | 0.00* | 0.00* | 0.00* | 0.00* | 0.00* | 0.00* |
| **Average** | – | **79.80** | 65.26 | 63.30 | 33.11 | 45.40 | 47.71 | **55.77** | 45.52 | 39.26 | 41.84 | 23.33 |
| **Conceptual Knowledge** | | | | | | | | | | | | |
| Social Science | VC | **83.00** | 77.00 | 72.00 | 37.25 | 32.75 | 75.75 | **76.50** | 63.25 | 62.00 | 54.00 | 15.75 |
| | VQ | **95.75** | 83.75 | 86.50 | 46.00 | 72.75 | 75.50 | **77.75** | 72.50 | 66.75 | 70.00 | 50.00 |
| | IF | **84.50** | 59.00 | 54.75 | 22.50 | 22.00 | 34.25 | **46.00** | 25.50 | 15.00 | 27.25 | 14.25 |
| | KP | **78.75** | 53.00 | 48.75 | 16.75 | 11.25 | 25.25 | **38.25** | 17.50 | 10.50 | 20.50 | 10.25 |
| | Avg | **85.50** | 68.19 | 65.50 | 30.63 | 34.69 | 52.69 | **59.63** | 44.69 | 38.56 | 42.94 | 22.56 |
| Natural Science | VC | **80.00** | 65.00 | 70.25 | 31.00 | 35.00 | 65.75 | 68.00 | **71.25** | 61.75 | 47.00 | 18.75 |
| | VQ | **96.00** | 83.75 | 87.25 | 47.00 | 75.50 | 76.00 | **80.25** | 78.00 | 77.75 | 72.75 | 58.25 |
| | IF | **76.50** | 44.75 | 48.00 | 18.25 | 25.00 | 38.25 | **49.00** | 27.50 | 18.25 | 19.00 | 17.50 |
| | KP | **67.75** | 34.25 | 39.25 | 12.50 | 18.25 | 28.00 | **40.25** | 19.50 | 14.00 | 13.50 | 11.75 |
| | Avg | **80.06** | 56.94 | 61.19 | 27.19 | 38.44 | 52.00 | **59.38** | 49.06 | 42.94 | 38.06 | 26.56 |
| **Average** | – | **81.37** | 59.65 | 62.23 | 28.02 | 37.54 | 52.17 | **59.44** | 48.01 | 41.88 | 39.24 | 25.59 |
| **Procedural Knowledge** | | | | | | | | | | | | |
| Logical Reasoning | VC | **81.00** | 73.50 | 64.75 | 15.00 | 23.50 | 74.75 | 71.25 | 58.75 | 55.50 | 37.25 | 14.75 |
| | VQ | **95.00** | 84.50 | 85.00 | 26.75 | 66.25 | 84.25 | 83.00 | 72.25 | 72.75 | 75.50 | 58.75 |
| | IF | **59.25** | 33.00 | 24.75 | 4.25 | 7.25 | 23.25 | **29.25** | 20.25 | 10.25 | 5.25 | 3.75 |
| | KP | **51.00** | 25.50 | 16.50 | 1.75 | 2.25 | 16.25 | **21.25** | 12.25 | 7.75 | 2.00 | 2.00 |
| | Avg | **71.56** | 54.13 | 47.75 | 11.94 | 24.81 | 49.63 | **51.19** | 40.88 | 36.56 | 30.00 | 19.81 |
| Instruction Decomposition | VC | **71.00** | 58.25 | 51.50 | 28.75 | 31.00 | 30.75* | **32.25*** | 25.75* | 29.75* | 20.75* | 9.50* |
| | VQ | **96.25** | 82.50 | 76.75 | 46.50 | **64.75** | 29.00* | 25.25* | 26.50* | 39.25* | 39.25* | 27.75* |
| | IF | **88.00** | 74.25 | 53.50 | 32.25 | **39.25** | 32.75* | 24.50* | 16.00* | 11.75* | 9.25* | 7.00* |
| | Avg | **85.08** | 71.67 | 60.58 | 35.83 | **45.00** | 30.83* | 27.33* | 22.75* | 26.92* | 23.08* | 14.75* |
| **Average** | – | **78.32** | 62.90 | 54.17 | 23.89 | 34.91 | **40.23** | 39.26 | 31.82 | 31.74 | 26.54 | 17.28 |
| **Overall Average** | | **80.09** | 62.41 | 60.70 | 28.85 | 39.70 | 47.76 | **53.36** | 43.29 | 38.55 | 37.15 | 22.82 |

## 4.2 Results and Analysis

**Overall Performance.** Table 2 reports evaluation results across various knowledge types, spanning seven dimensions with different metrics. All scores are normalized to a 100-point scale to enable straightforward comparison. The results reveal that closed-source models substantially outperform open-source models on KRIS-Bench. BAGEL-Think achieves the best performance among open-source models and has begun to approach the performance level of closed-source models such as Gemini 2.0 and Doubao. Notably, we observe that introducing a reasoning process into BAGEL (BAGEL-Think) yields a marked improvement over the baseline BAGEL model without reasoning, highlighting the critical role of reasoning in KRIS-Bench. Among all models, GPT-4o achieves the highest overall scores across nearly all knowledge types and evaluation dimensions, except for slightly lagging behind Gemini 2.0 in visual consistency for temporal prediction.

**Analysis by Knowledge Types.** Based on Table 2, nearly all models consistently perform the weakest on procedural knowledge, indicating significant challenges in multi-step reasoning and task decomposition for current editing models. Surprisingly, models do not consistently struggle more with conceptual knowledge than with factual knowledge, despite the former requiring a higher level of abstraction and generalization. In particular, models such as GPT-4o, BAGEL, BAGEL-Think, Step1X-Edit, and AnyEdit perform slightly worse on factual knowledge tasks than on conceptual ones. This counterintuitive finding suggests that the current strong image generation models still lack robust grounding in perceptual and real-world facts, such as object counting and spatial positioning.

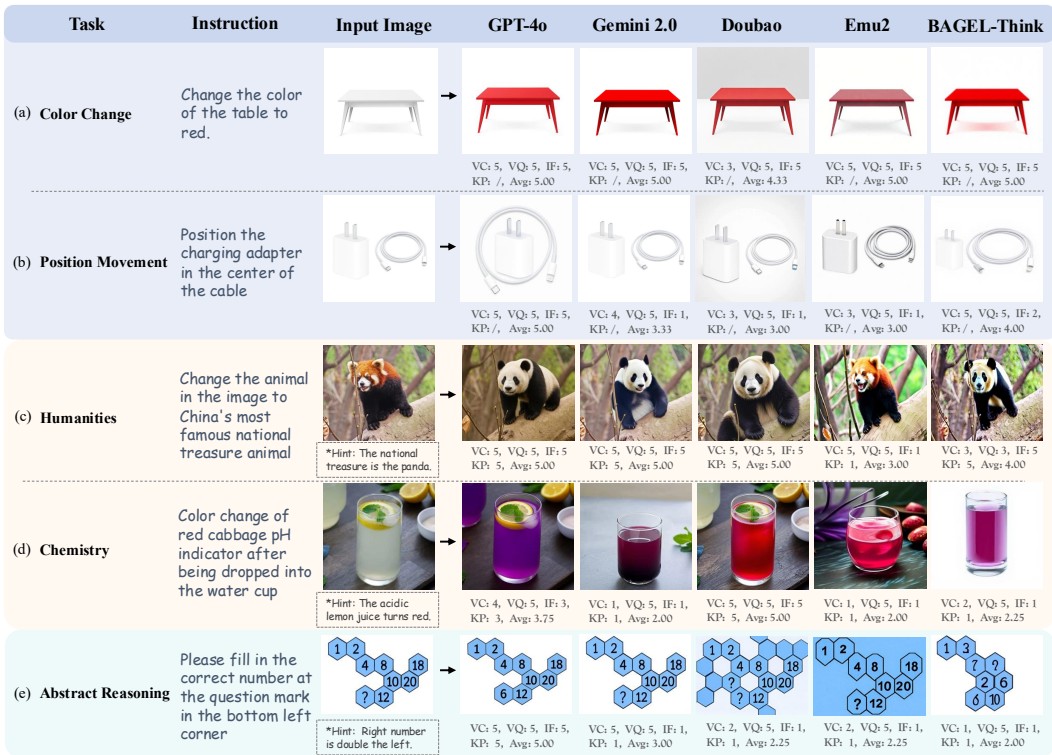

Figure 3: Visualization results of (a) Color Change, (b) Position Movement, (c) Humanities, (d) Chemistry, and (e) Abstract Reasoning across different models and metrics. Each example is provided with scores across the four evaluation metrics as well as an overall average score. Note that the knowledge hint is provided solely for evaluation and has been shortened for better illustration.

**Analysis by Reasoning Dimensions.** Within each knowledge type, a closer breakdown of reasoning dimensions reveals diverse performance patterns in Table 2. For factual knowledge, most models achieve relatively high accuracy in attribute-level perception tasks (Figures 3 (a)), but exhibit sharp drops in spatial reasoning (Figures 3 (b)). For conceptual knowledge, models generally perform better on tasks requiring commonsense or cultural knowledge, but struggle with tasks grounded in scientific principles where expert-domain reasoning is needed. As illustrated in Figure 3 (c–d), although the models demonstrate strong performance on the humanities task by correctly identifying the panda as China's most iconic national treasure, they exhibit significant limitations in scientific reasoning, such as failing to accurately interpret chemical reactions and overlooking the fact that red cabbage turns red in acidic conditions. For procedural knowledge, closed-source models exhibit significantly stronger performance on instruction decomposition tasks, with GPT-4o achieving particularly notable results. In contrast, all models face considerable challenges in logical reasoning tasks involving symbolic manipulation or abstract pattern recognition. Interestingly, GPT-4o occasionally succeeds in solving such tasks (Figure 3 (e), the value on the right is twice that of the value on the left), highlighting its emerging capacity for logical reasoning.

**Analysis by Editing Tasks and Metrics.** Figure 4 presents a radar chart depicting model performance across various editing tasks and metrics. The results reveal substantial variation in performance across specific tasks, even within the same reasoning dimension. For example, under the Attribute Perception category, both Gemini 2.0 and Doubao perform noticeably worse on Count Change and Size Adjustment compared to Color Change in terms of instruction following. Furthermore, while all models attain relatively high scores in Visual Consistency and Visual Quality, their performance in Instruction Following and Knowledge Plausibility exposes significant shortcomings. Notably, scores for Knowledge Plausibility are consistently lower than those for Instruction Following, highlighting persistent challenges in integrating and applying external knowledge accurately during editing. More-over, BAGEL-Think surpasses nearly all other open-source models on the Knowledge Plausibility metric across most tasks. Remarkably, it even outperforms closed-source models such as Gemini 2.0 and Doubao in Biology and Chemistry tasks.

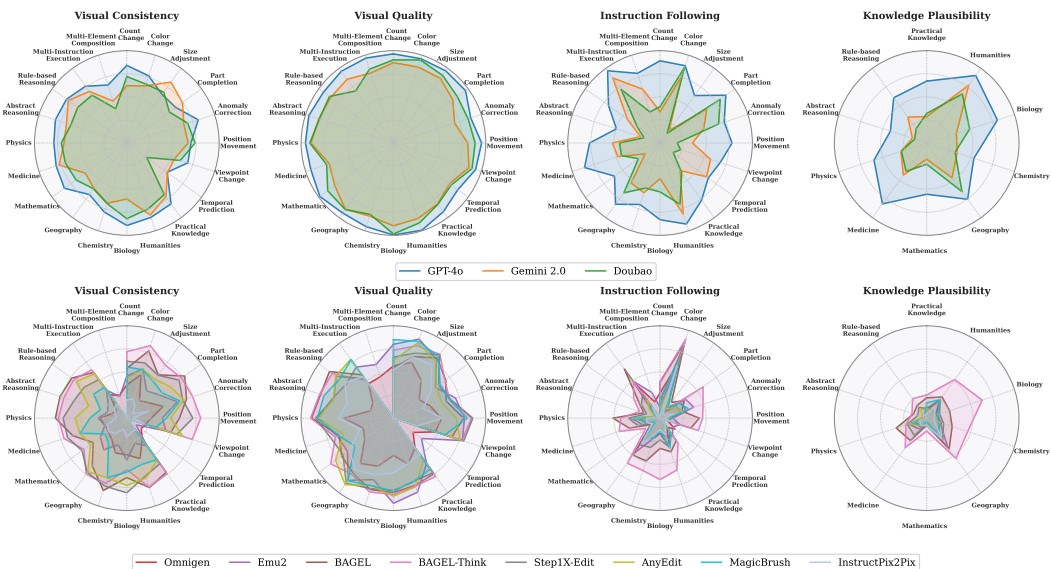

Figure 4: Performance on KRIS-Bench across different editing tasks and four different metrics. Top: closed-source models. Bottom: open-source models.

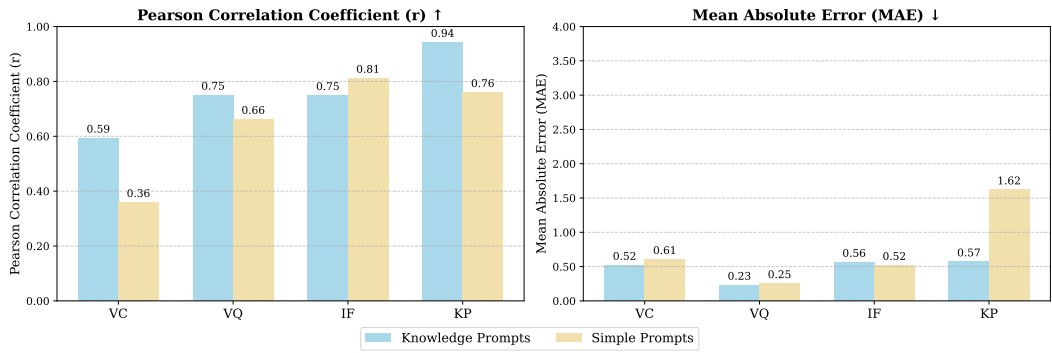

Figure 5: Correlation between human and VLM scores across Visual Consistency (VC), Visual Quality (VQ), Instruction Following (IF), and Knowledge Plausibility (KP). We compare the prompts incorporating knowledge hints (Knowledge Prompts) with a simple baseline (Simple Prompts).

These comprehensive analyses reveal that despite recent advancements in instruction-based image editing, current models exhibit inherent limitations in knowledge-centric reasoning. The challenges extend beyond the completion of complex edits to encompass the comprehension and application of diverse forms of knowledge in a coherent and grounded manner. By anchoring the evaluation on a cognitively informed taxonomy, KRIS-Bench surpasses task-specific benchmarks to systematically evaluate how models internalize, manipulate, and operationalize knowledge. This paradigm shift offers new pathways for developing editing models that engage in reasoning processes more analogous to human cognition. In addition, the performance gains observed in BAGEL-Think through the integration of a reasoning process on certain tasks suggest a promising direction for tackling knowledge-based reasoning challenges. Additional experimental results are provided in the Appendix.

### 4.3 Assessment of Evaluation Protocol

To evaluate the reliability of VLM scores, we recruited 12 human annotators, all with at least an undergraduate-level education, to conduct the user study, given that KRIS-Bench involves knowledge-based reasoning tasks. The study adhered to ethical standards, with compensation set above the local minimum wage. All annotators received at least one round of training and performed a trial annotation session. Their results were then reviewed and discussed in pairs to ensure alignment with the evaluation criteria. Given the potential subjectivity in human scoring, we normalized the raw scores into three qualitative categories: *Good*, *Fair*, and *Poor*, which were subsequently mapped to

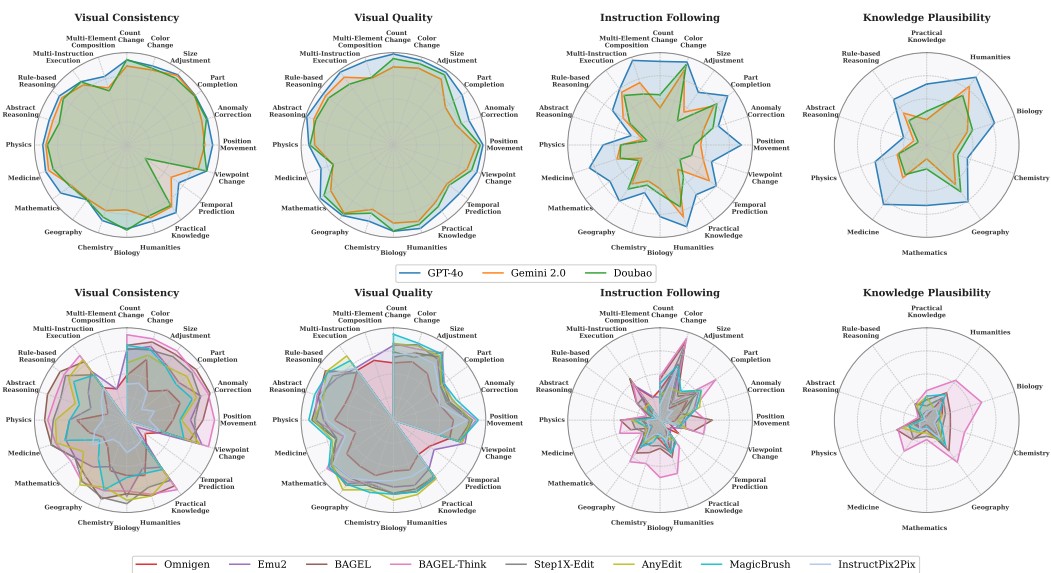

Figure 6: Performance on KRIS-Bench across different editing tasks and four different metrics using Qwen2.5-VL-72B as scoring VLM. Top: closed-source models. Bottom: open-source models.

numerical scores of 5, 3, and 1, respectively. For each sample, we collected ratings from at least two annotators, and the final score was computed as the average of the individual ratings.

We report the Pearson correlation coefficient ($r$) and Mean Absolute Error (MAE) between the expert ratings and the scores produced by the VLM, as shown in Figure 5. We compared our carefully designed prompts incorporating knowledge hints (Knowledge Prompts) with a simple baseline (Simple Prompts). The results show that Knowledge Prompts yield stronger $r$ and lower MAE values, especially for the Knowledge Plausibility metric. This indicates that our knowledge-enhanced prompts provide more accurate evaluations for knowledge-based reasoning in image editing. All scoring prompts are provided in the Appendix.

## 4.4 Open-source VLM Evaluation

To ensure transparency and reproducibility, we adopt the open-source vision-language model Qwen2.5-VL-72B as a proxy judge to score the predictions of each evaluated model. The results are presented in Figure 6. As shown, the scoring trends across tasks align closely with those obtained using GPT-4o (May 2025) in Figure 4. Table 3 further summarizes the performance across different knowledge dimensions and evaluation metrics based on Qwen2.5-VL-72B's assessments.

## 5 Conclusion

We introduce KRIS-Bench, a cognitively grounded benchmark designed to systematically evaluate the reasoning capabilities of image editing models through the lens of factual, conceptual, and procedural knowledge. In contrast to prior task-oriented or content-driven benchmarks, KRIS-Bench establishes a knowledge-centric framework that integrates fine-grained task categorization with human-calibrated evaluation protocols, enabling a more interpretable and diagnostic understanding of model reasoning behaviors. Our empirical results reveal persistent and systematic gaps in current models' ability to reason across diverse knowledge types, underscoring the need for deeper cognitive integration and more balanced reasoning supervision in future image editing systems.

**Limitations.** While KRIS-Bench represents a comprehensive attempt to construct a knowledge-based reasoning image editing benchmark with broader task coverage and richer evaluation dimensions than existing alternatives, it is not without limitations. Potential issues include the relatively modest dataset scale, uneven distribution across knowledge categories, and cultural or contextual biases embedded in task design. Future extensions may address these challenges through larger-scale, cross-cultural data collection and iterative human validation.

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

# Supplementary Material

## A  Detailed Tasks Explanation

Based on the previously defined knowledge categories, we further refine them into 7 capability dimensions, each capturing a distinct aspect of visual reasoning. To systematically evaluate these dimensions, we design a suite of 22 representative tasks that span a wide range of perceptual, conceptual, and procedural challenges. In the following section, we comprehensively explain each tasks.

### A.1  Factual Knowledge

Tasks in this category evaluate fundamental visual and temporal understanding that does not require external knowledge or abstract reasoning. These tasks rely on direct perception and low-level cognitive operations. We divide this category into three sub-dimensions: Attribute Perception, Spatial Perception, and Temporal Prediction.

**Attribute Perception:**

- **Count Change.** Modify the number of specific objects in an image based on the instruction, testing the model's ability to perceive and edit object quantities accurately.

- **Color Change.** Modify the color of a specified object or region, evaluating the model's ability to recognize and apply precise color transformations.

- **Size Adjustment.** Modify the size of a target object to match a reference, evaluating the model's understanding of relative scale and spatial consistency.

- **Part Completion.** Fill in missing or occluded parts of objects using visual context, testing spatial reasoning and shape completion ability.

- **Anomaly Correction.** Detect and fix visually or logically implausible elements—such as anatomical errors, structural anomalies, or impossible object configurations, to ensure real-world plausibility and visual coherence.

**Spatial Perception:**

- **Position Movement.** Move objects to target locations within the image, requiring spatial understanding and coherent object placement relative to surrounding elements.

- **Viewpoint Change.** Translate between different viewpoints (e.g., front, side, top) of the same object, testing spatial imagination and 3D reasoning ability.

**Temporal Prediction:**

- **Reverse Prediction.** Given several consecutive future frames, infer and reconstruct a plausible earlier frame in the sequence. This task tests the model's ability to reason backward over temporal dynamics while preserving consistency in motion and appearance.

- **Intermediate Prediction.** Predict a missing intermediate frame given the surrounding frames in a temporal sequence. This task requires understanding temporal continuity, motion interpolation, and visual coherence across multiple time steps.

- **Forward Prediction.** Predict the future frame based on several earlier frames in a visual sequence. This evaluates the model's ability to extrapolate motion and anticipate changes in the scene based on past observations.

### A.2  Conceptual Knowledge

Tasks in this category require understanding and applying real-world knowledge beyond perceptual cues. They often involve reasoning grounded in external knowledge systems, such as cultural norms, scientific principles, or domain-specific rules. We divide this category into two sub-dimensions: Social Science and Natural Science.

**Social Science:**

- **Practical Knowledge.** Apply everyday commonsense reasoning to adjust objects or scenarios in plausible, real-world ways, e.g., modifying a clock for daylight saving time or removing meat from a vegetarian meal.
- **Humanities.** Edit images based on cultural, historical, or religious context. Tasks require understanding symbolic elements such as traditional foods, attire, landmarks, or artifacts. For example, replacing a dish with mooncakes for the Mid-Autumn Festival.

**Natural Science:**

- **Biology.** Apply biological principles to depict realistic life stages, behaviors, or environmental responses, e.g., fruit ripening, animal defense reactions, or plant seasonal changes.
- **Chemistry.** Modify images based on chemical properties, reactions, or material transformations. For example, show color changes from pH indicators or gas generation during acid–base reactions.
- **Geography.** Modify images by incorporating spatial, climatic, and geological concepts. This includes changes in terrain, celestial events, or weather-related effects such as snowfall, tides, or desertification.
- **Mathematics.** Perform modifications guided by mathematical concepts, including geometric properties, algebraic transformations, graph theory, and so on.
- **Medicine.** Apply medical understanding to visualize anatomical structure, physiological signals, pathological symptoms, or treatment-related conditions.
- **Physics.** Apply knowledge of physical laws and principles such as motion, force, thermodynamics, optics, and electromagnetism to guide image modifications.

### A.3 Procedural Knowledge

Tasks in this category involve executing structured reasoning processes and following complex or multi-step instructions that go beyond simple visual matching. These tasks typically require planning, rule-following, and the integration of multiple operations into a coherent output. We divide this category into two sub-dimensions: Logical Reasoning and Instruction Decomposition.

**Logical Reasoning:**

- **Abstract Reasoning.** Reason about symbolic structures, numerical relationships, or high-level conceptual patterns that go beyond literal visual interpretation, often requiring logical deduction, analogy, or transformation rules.
- **Rule-based Reasoning.** Apply explicit and well-defined rules to guide visual transformations, such as maze solving, game logic (e.g., Sudoku, Tic-Tac-Toe), or constraint satisfaction, requiring precise adherence to task constraints and rule consistency.

**Instruction Decomposition:**

- **Multi-instruction Execution.** This category focuses on executing multiple sequential editing instructions in a coherent manner. A typical task involves designing posters or product visuals from a given object, requiring identity preservation and edits such as background generation, text placement, and lighting adjustment.
- **Multi-element Composition.** This category focuses on integrating visual elements from multiple sources into a coherent scene. Representative tasks include replacing clothing with a provided reference or inserting objects from several images, requiring segmentation, spatial reasoning, and consistent visual blending.

## B Data Distribution

To support a comprehensive evaluation of knowledge-based image editing, our benchmark comprises a total of 1,267 instances spanning 22 task types. Each task is designed to reflect a unique combination of knowledge requirements and reasoning dimensions. Figure 7 shows three views of the dataset: by knowledge type (left), by reasoning dimension (center), and by individual editing task (right).

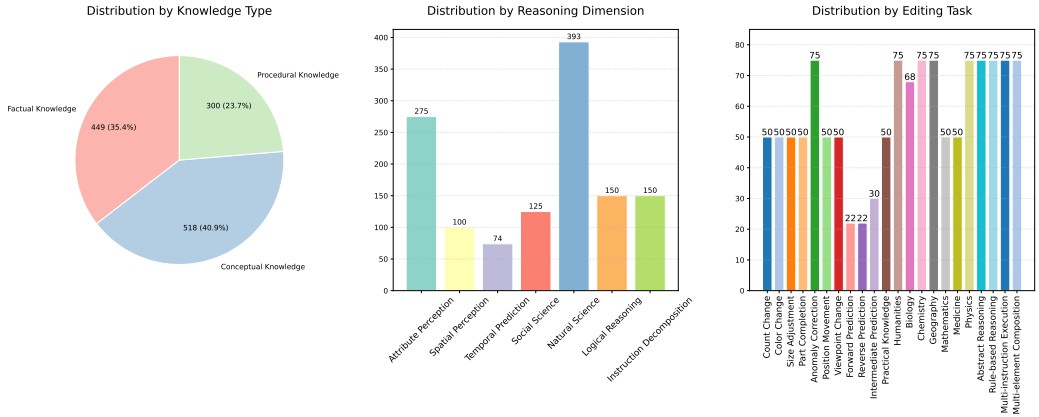

Figure 7: Distribution of KRIS-Bench instances by knowledge type (left), reasoning dimension (center), and editing task (right).

**Knowledge Type Breakdown.** Conceptual Knowledge has the most instances (518, 40.9%), followed by Factual Knowledge (449, 35.4%) and Procedural Knowledge (300, 23.7%).

**Reasoning Dimension Breakdown.** Natural Science dominates with 393 instances (31.0%), followed by Attribute Perception (275, 21.7%). Logical Reasoning and Instruction Decomposition each contribute 150 (11.8%), with Social Science (125, 9.9%), Spatial Perception (100, 7.9%), and Temporal Prediction (74, 5.8%) trailing behind.

**Editing Task Breakdown.** Among 22 unique tasks, nine have the highest count of 75 (5.9%), including Mathematics, Abstract Reasoning, and Multi-instruction Execution. Biology appears 68 times (5.4%), while perceptual tasks like Color Change and Size Adjustment each have 50 (3.9%).

## C  Computing Source Requirements

All experiments on open-source models were conducted on a server equipped with dual Intel Xeon Platinum 8468 CPUs (192 threads), 960 GB RAM, and 8×NVIDIA H100 80GB GPUs. Each model required approximately 2 hours to complete all 1,267 editing tasks. Closed-source models were accessed via official APIs or web platforms, where compute details are not user-controllable. No additional large-scale pretraining or auxiliary runs were performed beyond the reported experiments.

## D  Data Collection

Most images in our benchmark were collected from the internet under Creative Commons licenses to ensure eligibility for academic use. A smaller portion was generated using generative models [54] or sourced from existing datasets [13, 67–72]. For the *Viewpoint Change* task, we utilized 3D assets from the Amazon-Berkeley Objects (ABO) dataset [67] and Sketchfab (`https://sketchfab.com/`) to enable accurate evaluation with ground truth views. The *Abstract Reasoning* task includes atomic examples derived from prior works [68, 69] and extended through manual annotation. Some samples for the *Multi-element Composition* task were taken from virtual try-on datasets [70, 71]. For the *Temporal Prediction* dimension, we incorporated some clips from video object segmentation datasets [72] and searched through the internet, including freely available videos that permit academic use.

## E  More Visualization Results

In this section, we present additional qualitative results. The results show that most models struggle with Count Change tasks and often fail to correct anomalies in the image (Figure 8, Figure 9). For the Part Completion task, many models are unable to infer missing components in the image unless explicitly instructed (e.g., "complete the bottle cap") (Figure 10). In contrast, performance on the

Table 3: Performance of different models across different reasoning dimensions and metrics, including Visual Consistency (VC), Visual Quality (VQ), Instruction Following (IF), and Knowledge Plausibility (KP). Scores marked with ∗ indicate models unable to handle multi-image input tasks, with the corresponding task scores set to 0. The performance of open-source and closed-source models is separately marked with the best performance in **bold**, and the second best underlined. In this table, we use Qwen2.5-VL-72B as scoring VLM.

| Reasoning Dimension | Metric | Closed-Source Models | | | Open-Source Models | | | | | | | |
|---|---|---|---|---|---|---|---|---|---|---|---|---|
| | | GPT-4o | Gemini 2.0 | Doubao | OmniGen | Emu2 | BAGEL | BAGEL-Think | Step1X-Edit | AnyEdit | MagicBrush | InsPix2Pix |
| *Factual Knowledge* — Attribute Perception | VC | **91.75** | 88.00 | 90.25 | 54.25 | 77.50 | 89.25 | **92.00** | 82.50 | 73.50 | 74.75 | 34.00 |
| | VQ | **94.00** | 80.00 | 87.50 | 59.00 | 74.50 | 72.00 | 78.00 | 72.50 | 77.50 | **81.50** | 67.00 |
| | IF | **82.50** | 58.00 | 63.00 | 34.50 | 41.25 | 47.50 | **55.25** | 40.00 | 40.00 | 47.50 | 22.50 |
| | Avg | **89.42** | 75.33 | 80.25 | 49.25 | 64.42 | 69.58 | **75.08** | 65.00 | 63.58 | 67.92 | 41.17 |
| Spatial Perception | VC | **92.00** | 83.00 | 87.00 | 46.50 | 77.25 | 77.00 | **94.00** | 76.50 | 71.25 | 61.25 | 30.75 |
| | VQ | **96.00** | 87.00 | 91.50 | 64.50 | 81.75 | 79.25 | 82.25 | 81.75 | 81.00 | **83.50** | 65.75 |
| | IF | **73.25** | 46.50 | 36.50 | 13.75 | 28.25 | 47.50 | **49.50** | 23.75 | 24.50 | 25.25 | 13.50 |
| | Avg | **87.08** | 72.17 | 71.67 | 41.58 | 62.42 | 67.83 | **75.25** | 60.67 | 58.92 | 56.67 | 36.67 |
| Temporal Prediction | VC | **69.50** | 59.50 | 24.75 | **24.75** | 12.75 | 0.00* | 0.00* | 0.00* | 0.00* | 0.00* | 0.00* |
| | VQ | **88.75** | 71.50 | 74.25 | 48.25 | **54.50** | 0.00* | 0.00* | 0.00* | 0.00* | 0.00* | 0.00* |
| | IF | **75.25** | 66.00 | 26.75 | **24.75** | 13.75 | 0.00* | 0.00* | 0.00* | 0.00* | 0.00* | 0.00* |
| | Avg | **77.83** | 65.67 | 41.92 | **32.58** | 27.00 | 0.00* | 0.00* | 0.00* | 0.00* | 0.00* | 0.00* |
| **Average** | – | **86.99** | 73.03 | 72.02 | 44.79 | 57.81 | 57.73 | **62.75** | 53.32 | 52.06 | 54.22 | 33.38 |
| *Conceptual Knowledge* — Social Science | VC | **88.75** | 82.75 | 80.75 | 49.50 | 59.00 | 87.00 | **88.75** | 78.50 | 81.50 | 63.25 | 28.75 |
| | VQ | **91.75** | 82.00 | 86.50 | 51.25 | 70.00 | 76.75 | 79.75 | 77.75 | **81.50** | 79.00 | 63.25 |
| | IF | **79.75** | 57.25 | 56.75 | 22.50 | 18.50 | 34.75 | **48.00** | 28.75 | 23.50 | 33.75 | 19.25 |
| | KP | **78.25** | 53.00 | 51.50 | 16.50 | 12.50 | 29.25 | **42.75** | 22.75 | 20.00 | 30.25 | 12.75 |
| | Avg | **84.63** | 68.75 | 68.88 | 34.94 | 40.00 | 56.94 | **64.81** | 51.94 | 51.63 | 51.56 | 31.00 |
| Natural Science | VC | **87.00** | 78.00 | 82.25 | 47.00 | 66.50 | 81.50 | 80.25 | **82.75** | 77.50 | 60.50 | 32.50 |
| | VQ | **91.00** | 81.25 | 85.75 | 58.25 | 75.75 | 78.00 | 77.75 | 79.00 | **83.00** | 83.00 | 69.75 |
| | IF | **69.25** | 42.75 | 45.75 | 18.25 | 21.50 | 33.75 | **46.25** | 28.00 | 22.75 | 22.25 | 16.75 |
| | KP | **67.25** | 37.00 | 41.25 | 12.50 | 16.00 | 27.50 | **42.75** | 21.00 | 19.75 | 18.00 | 12.75 |
| | Avg | **78.63** | 59.75 | 63.75 | 34.00 | 44.94 | 55.19 | **61.75** | 52.69 | 50.75 | 45.94 | 32.94 |
| **Average** | – | **80.08** | 61.92 | 64.99 | 34.23 | 43.75 | 55.61 | **62.49** | 52.51 | 50.96 | 47.30 | 32.47 |
| *Procedural Knowledge* — Logical Reasoning | VC | **89.25** | 85.75 | 81.00 | 24.25 | 46.25 | **89.75** | 82.75 | 78.75 | 68.25 | 62.50 | 31.00 |
| | VQ | **96.25** | 90.75 | 87.50 | 55.50 | 81.00 | 88.00 | 87.75 | 81.25 | 84.50 | **88.75** | 84.00 |
| | IF | **48.25** | 34.50 | 27.75 | 5.00 | 8.00 | 20.75 | **23.00** | 20.00 | 12.00 | 11.25 | 5.50 |
| | KP | **43.75** | 28.75 | 21.25 | 1.50 | 4.00 | 13.25 | **15.25** | 14.00 | 11.00 | 8.25 | 3.50 |
| | Avg | **69.38** | 59.94 | 54.38 | 21.56 | 34.81 | **52.94** | 52.19 | 48.50 | 43.94 | 42.69 | 31.00 |
| Instruction Decomposition | VC | **81.25** | 72.75 | 73.00 | 39.50 | 50.25 | 39.25* | **43.25*** | 30.25* | 40.50* | 30.25* | 21.50* |
| | VQ | **96.75** | 83.25 | 79.25 | 67.25 | 70.50 | 31.75* | 31.50* | 32.25* | **43.00*** | 39.75* | 34.75* |
| | IF | **85.50** | 70.75 | 62.25 | 34.75 | 36.25 | 27.75* | **25.25*** | 15.25* | 10.75* | 9.50* | 5.75* |
| | Avg | **87.83** | 75.58 | 71.50 | 47.17 | **52.33** | 32.92* | 33.33* | 25.92* | 31.42* | 26.50* | 20.67* |
| **Average** | – | **78.61** | 67.76 | 62.94 | 34.37 | **43.57** | 42.93 | 42.76 | 37.21 | 37.68 | 34.60 | 25.84 |
| **Overall Average** | | **82.18** | 67.24 | 67.00 | 38.00 | 48.69 | 53.36 | **57.91** | 49.17 | 48.21 | 46.74 | 31.22 |

Color Change task is generally strong across all models (Figure 11). In the Spatial Perception dimension, GPT-4o consistently outperforms other models, especially in tasks involving Viewpoint Change and Position Movement (Figure 13, Figure 14). However, its performance on the Size Adjustment task is relatively weak, frequently failing to apply the correct edits (Figure 12). Regarding Temporal Prediction, both GPT-4o and Gemini 2.0 demonstrate a certain degree of temporal reasoning with logically coherent outputs. In contrast, models such as Doubao, OminiGen, and Emu2 generally fail to generate reasonable predictions (Figure 15).

We further present results on Conceptual Knowledge across multiple domains in Figures 16, 17, 18, 19, 20, 21, 22, and 23. Open-source models rarely succeed on these tasks, possibly due to the domain-specific nature of the content, which may fall outside their training distributions.

Interestingly, all three closed-source models exhibit some capability in Instruction Decomposition (Figure 24, Figure 25). However, they fall short in the Logical Reasoning dimension (Figure 26, Figure 27), highlighting significant limitations in current models' logical reasoning abilities.

# F  Evaluation Prompts

Figures 28, 29, and 30 illustrate the prompts used to evaluate Visual Consistency, Visual Quality, and Instruction Following, respectively. Specifically, for the reasoning dimension involving Knowledge Plausibility, we observed that evaluating Instruction Following and Knowledge Plausibility separately can introduce inconsistencies and lead to inaccurate model assessments. Thus, we jointly evaluate both aspects in a single prompt, as shown in Figures 31 and 32. Considering that Temporal Prediction and Multi-element Composition involve multiple reference images, we designed customized prompts

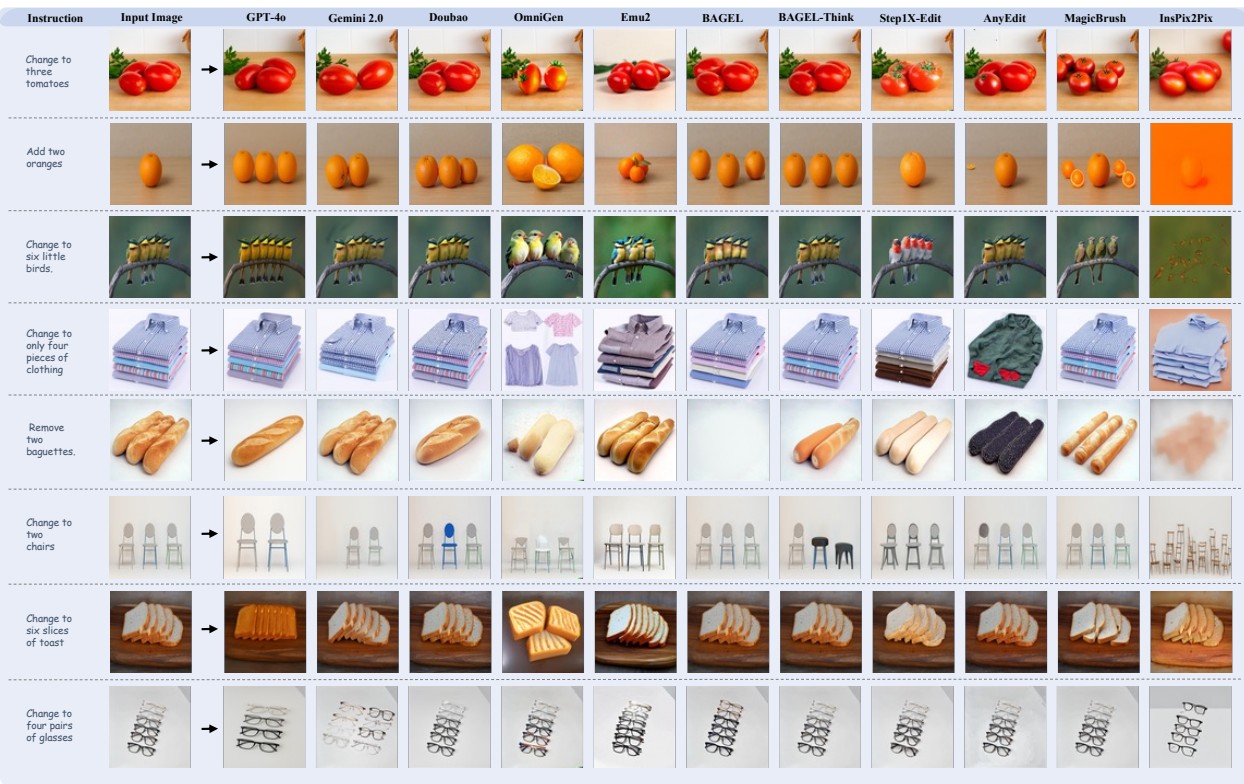

Figure 8: Visualization results of Count Change task.

for evaluating Visual Consistency and Instruction Following, presented in Figures 33 and 34. For the Viewpoint Change task, where ground truth images are available, we provide an additional Instruction Following prompt that uses the ground truth image as a reference, shown in Figure 35. As shown in Figure 36, we design a dedicated prompt for the Anomaly Correction task by incorporating a knowledge hint to facilitate accurate evaluation.

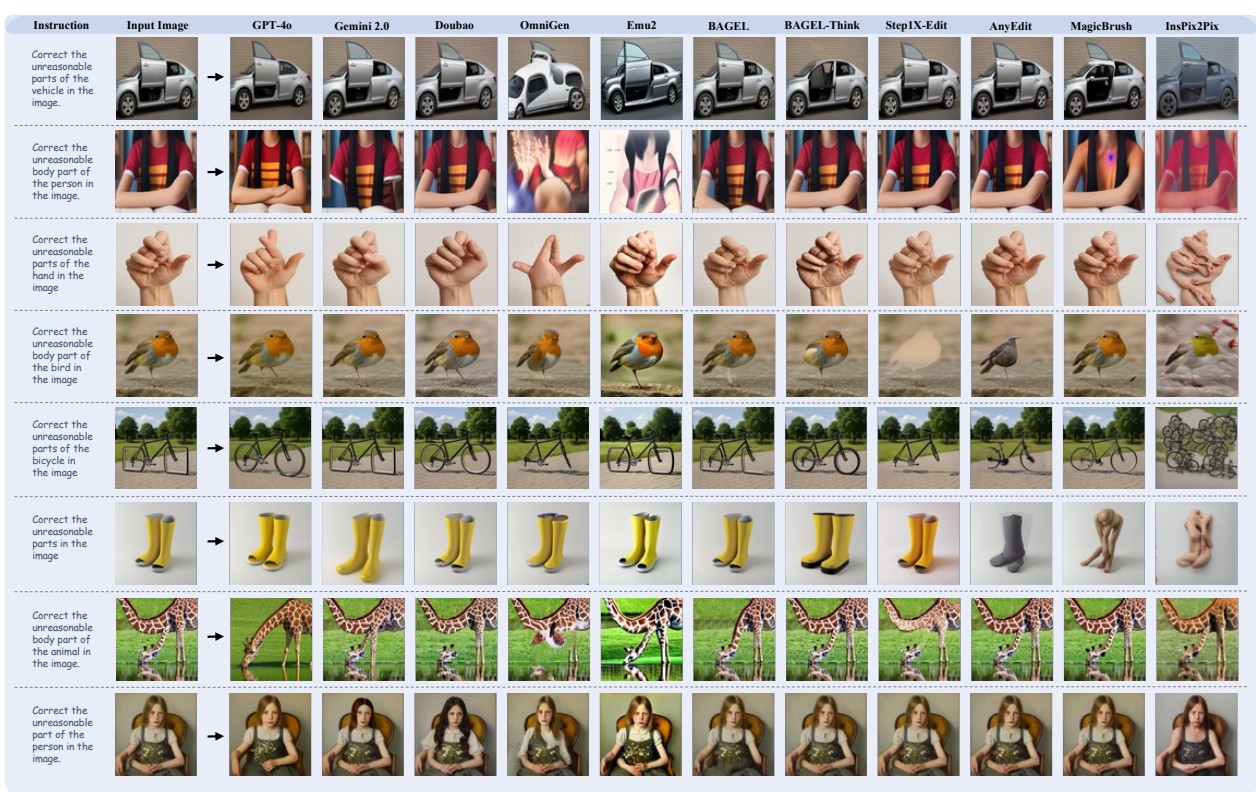

Figure 9: Visualization results of Anomaly Correction task.

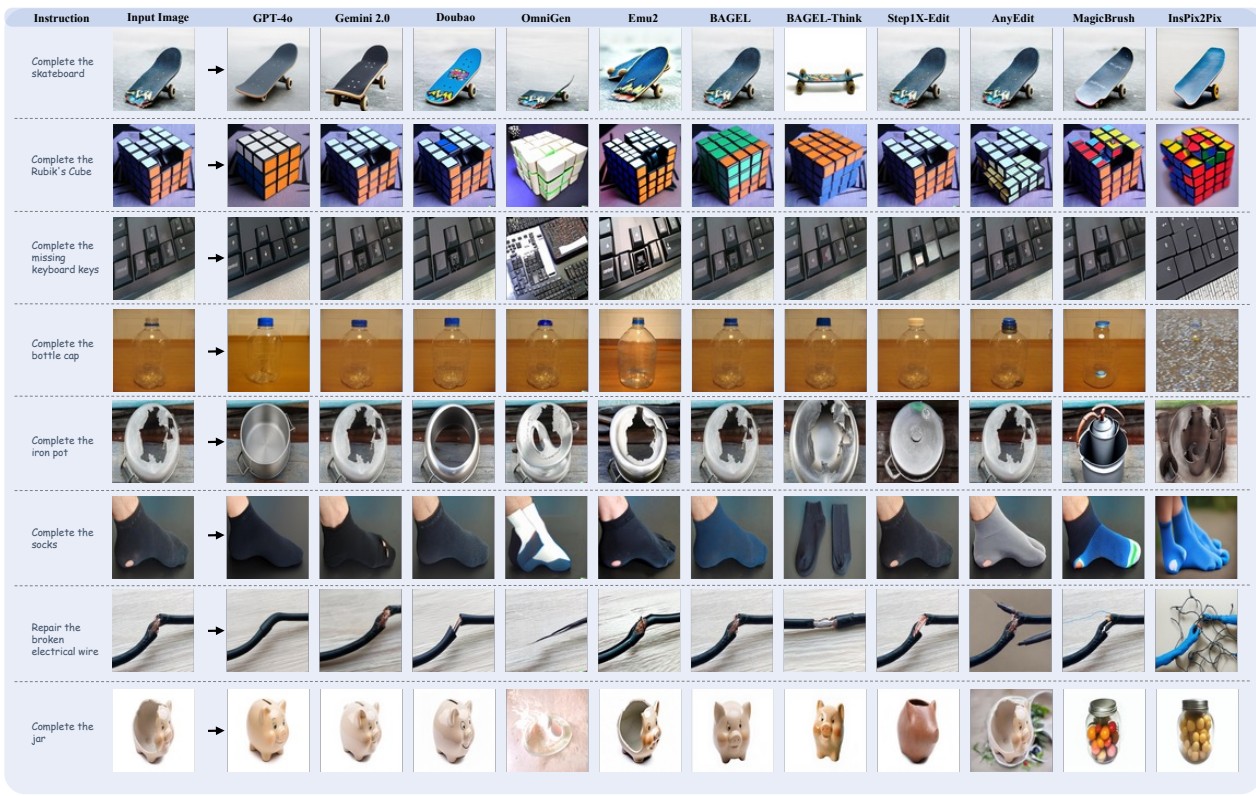

Figure 10: Visualization results of Part Completion task.

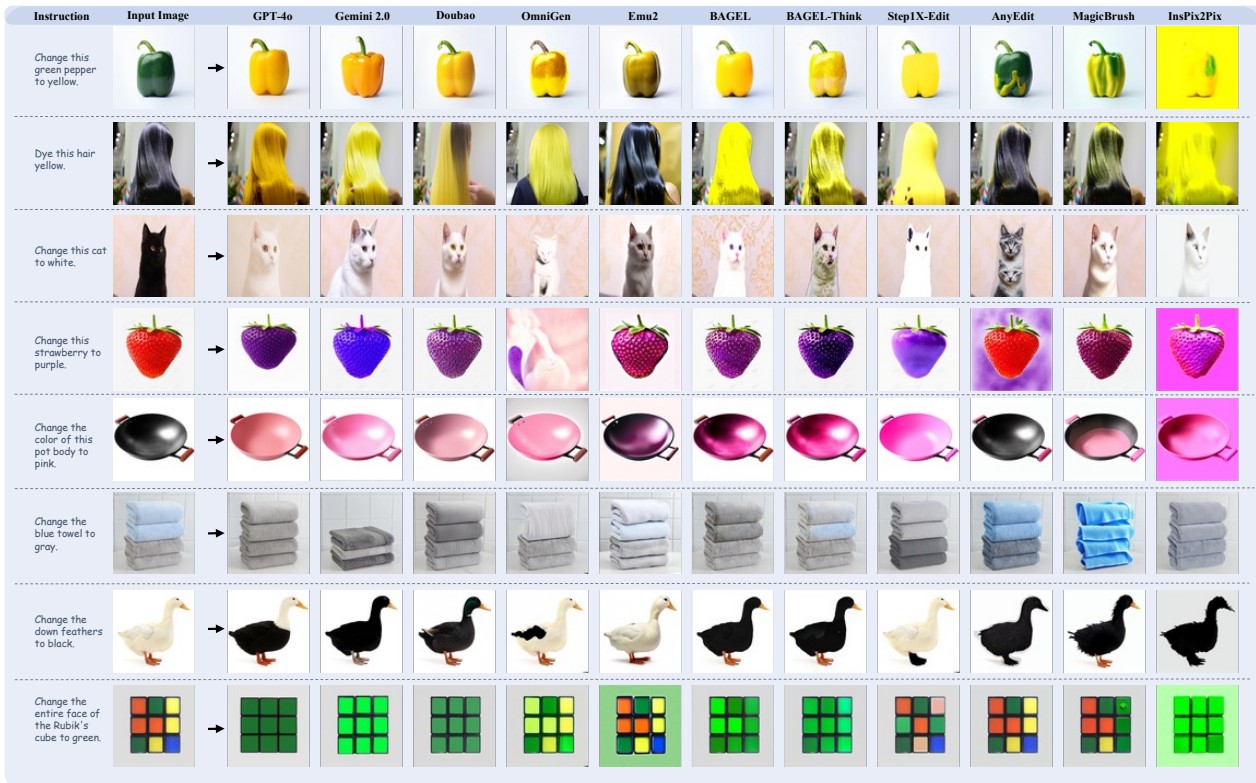

Figure 11: Visualization results of Color Change task.

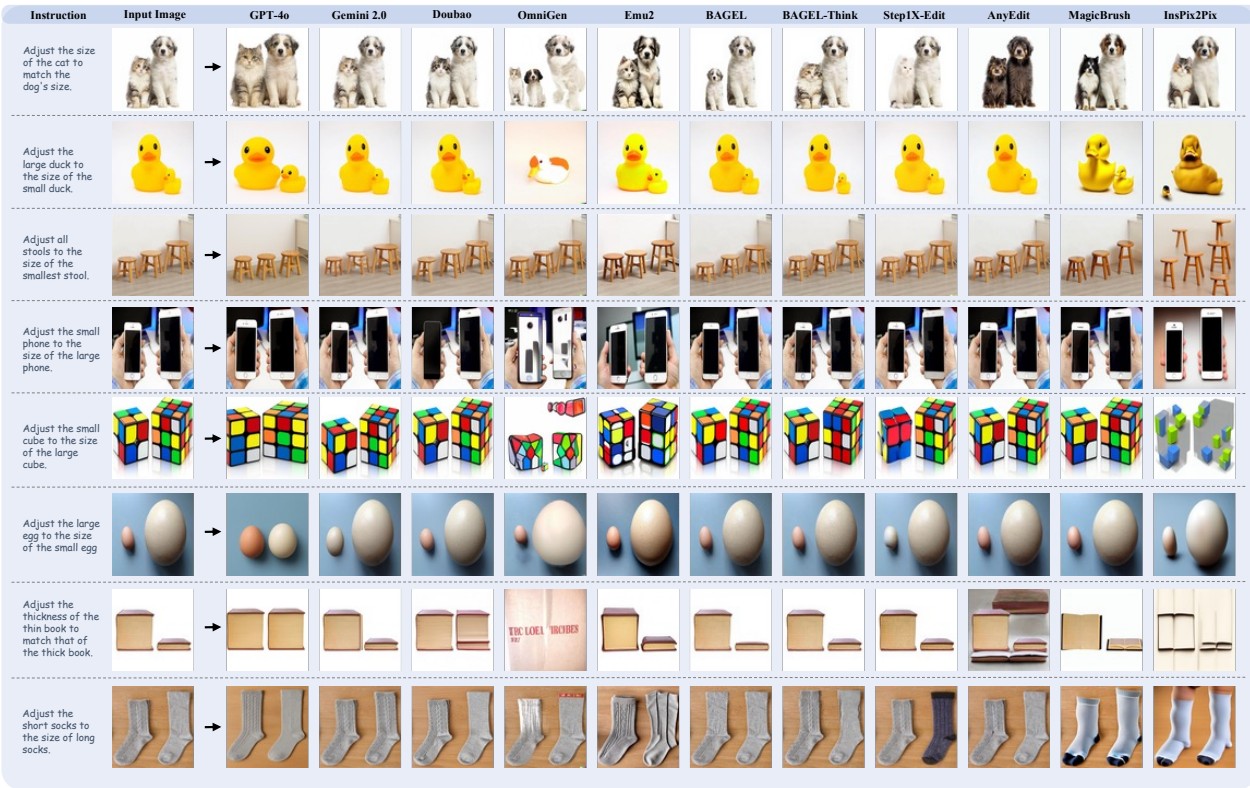

Figure 12: Visualization results of Size Adjustment task.

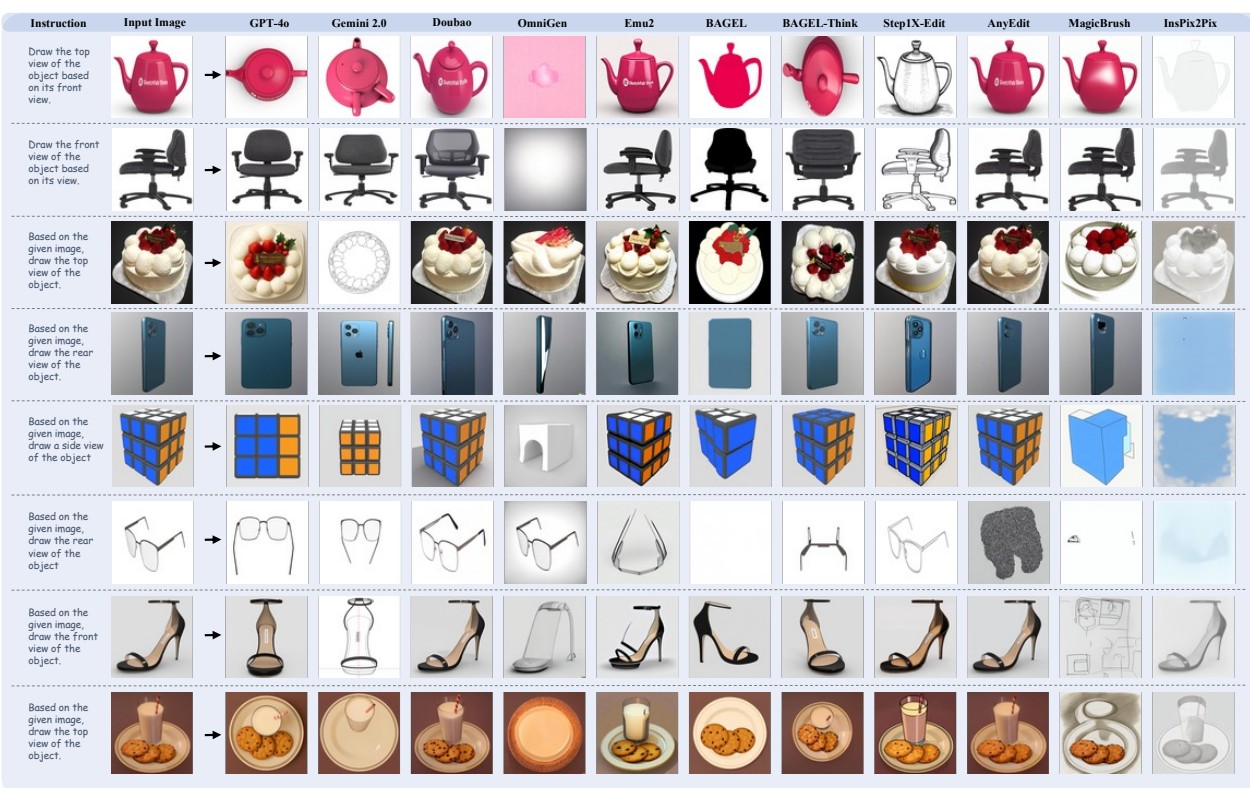

Figure 13: Visualization results of Viewpoint Change task.

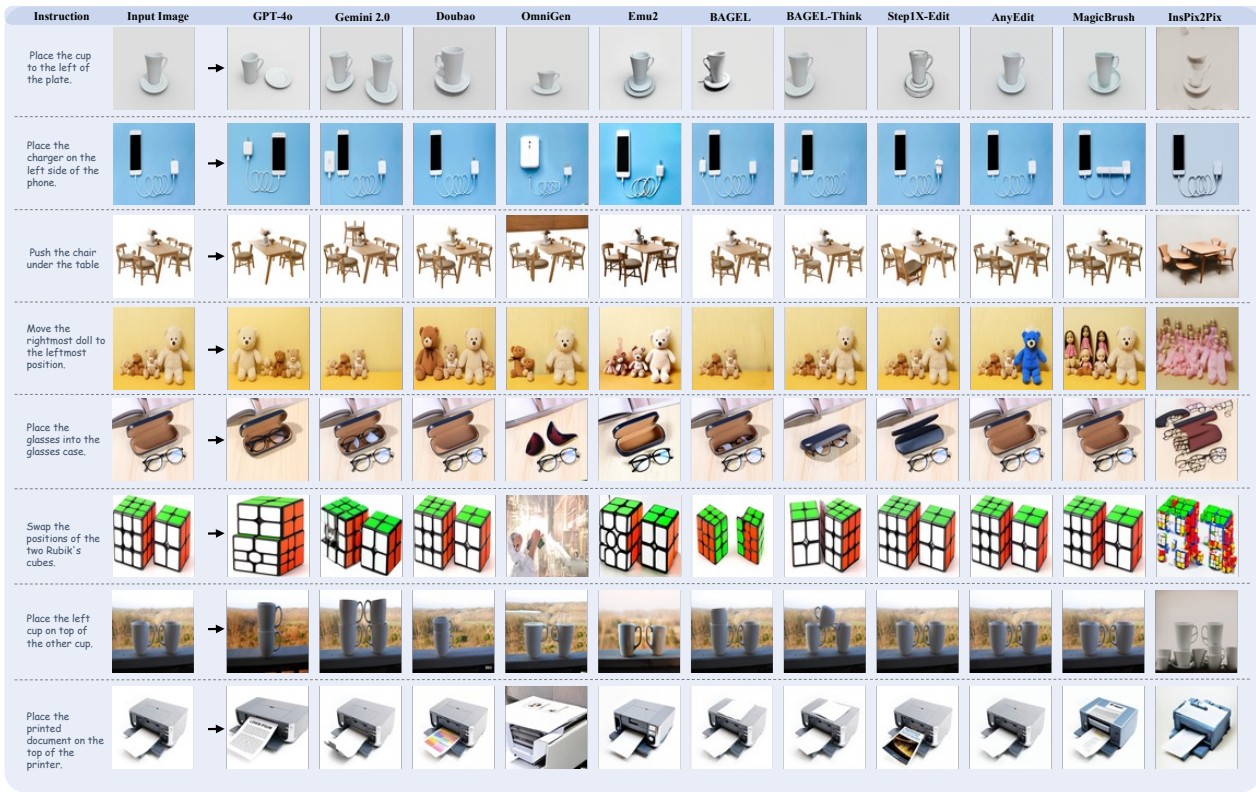

Figure 14: Visualization results of Position Movement task.

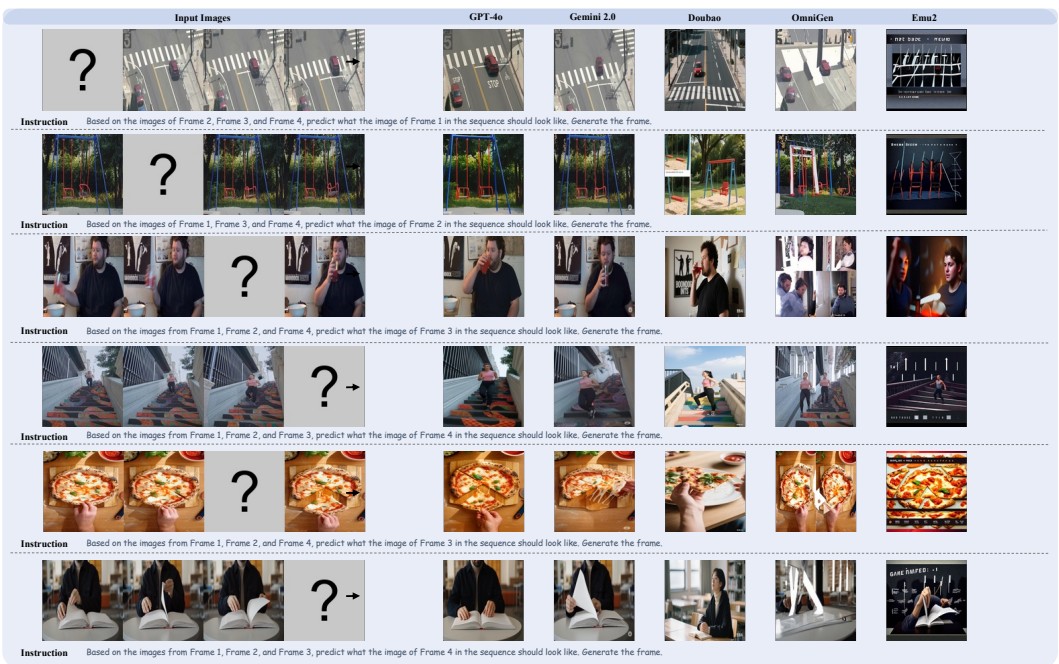

Figure 15: Visualization results of Temporal Prediction tasks.

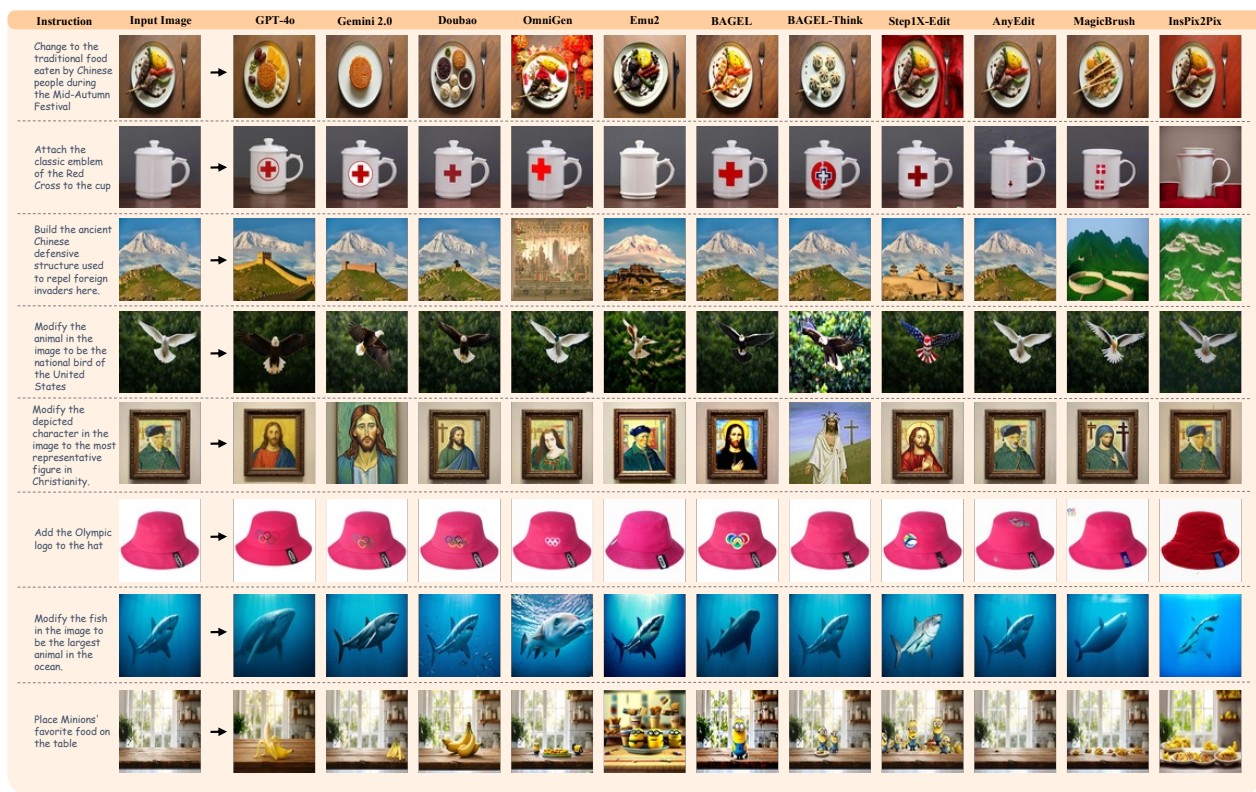

Figure 16: Visualization results of Humanities task.

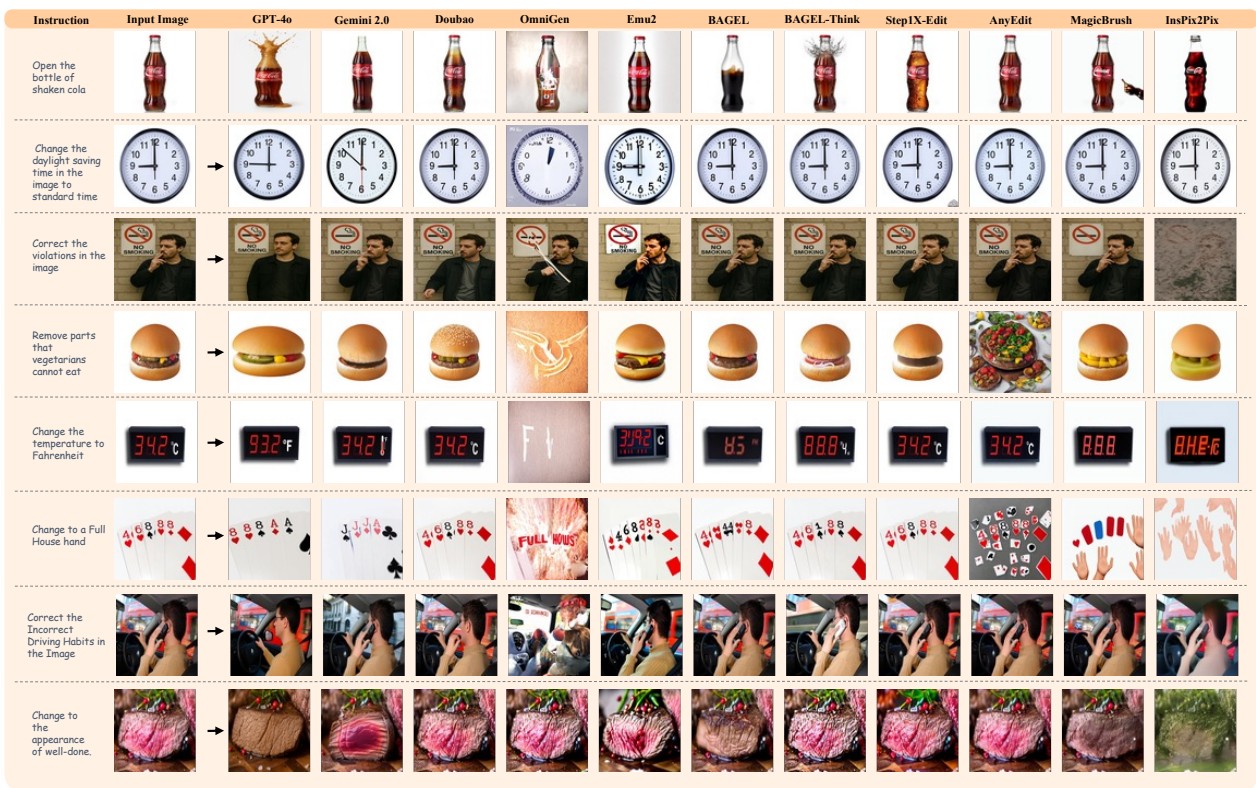

Figure 17: Visualization results of Practical Knowledge task.

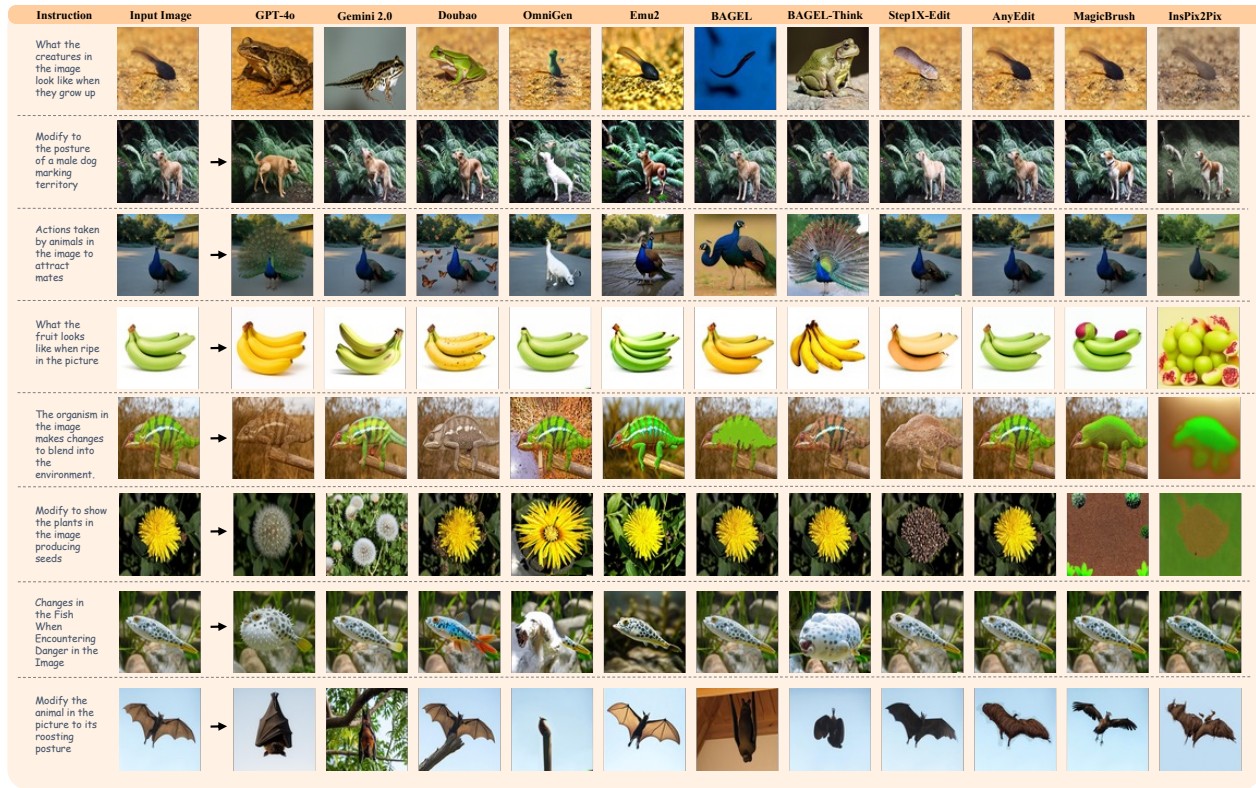

Figure 18: Visualization results of Biology task.

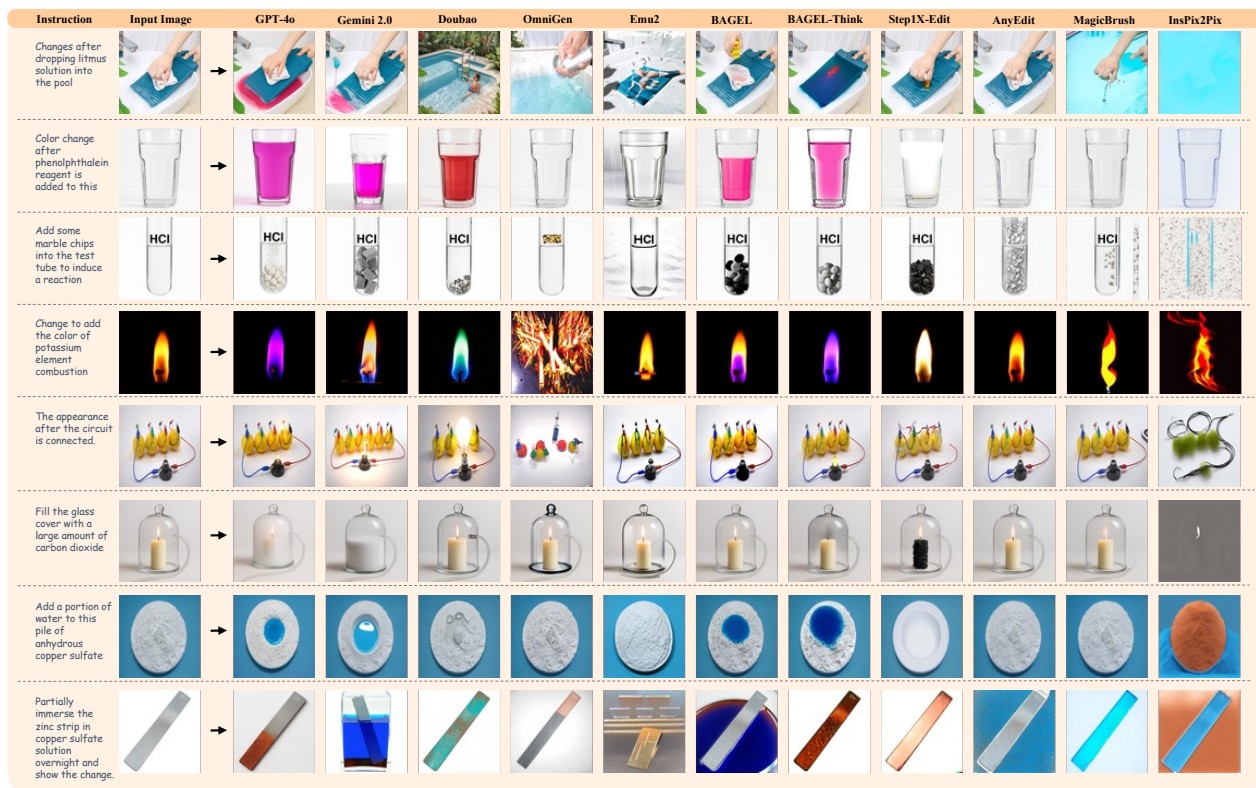

Figure 19: Visualization results of Chemistry task.

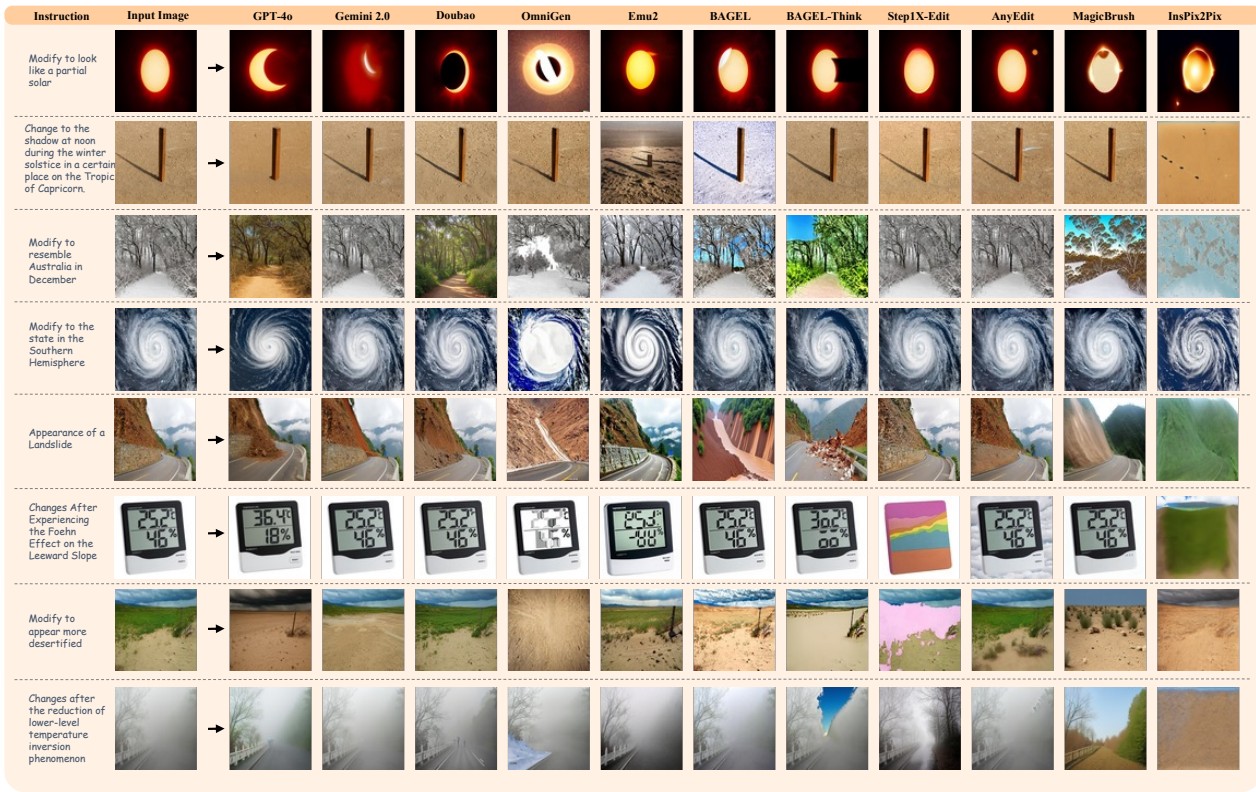

Figure 20: Visualization results of Geography task.

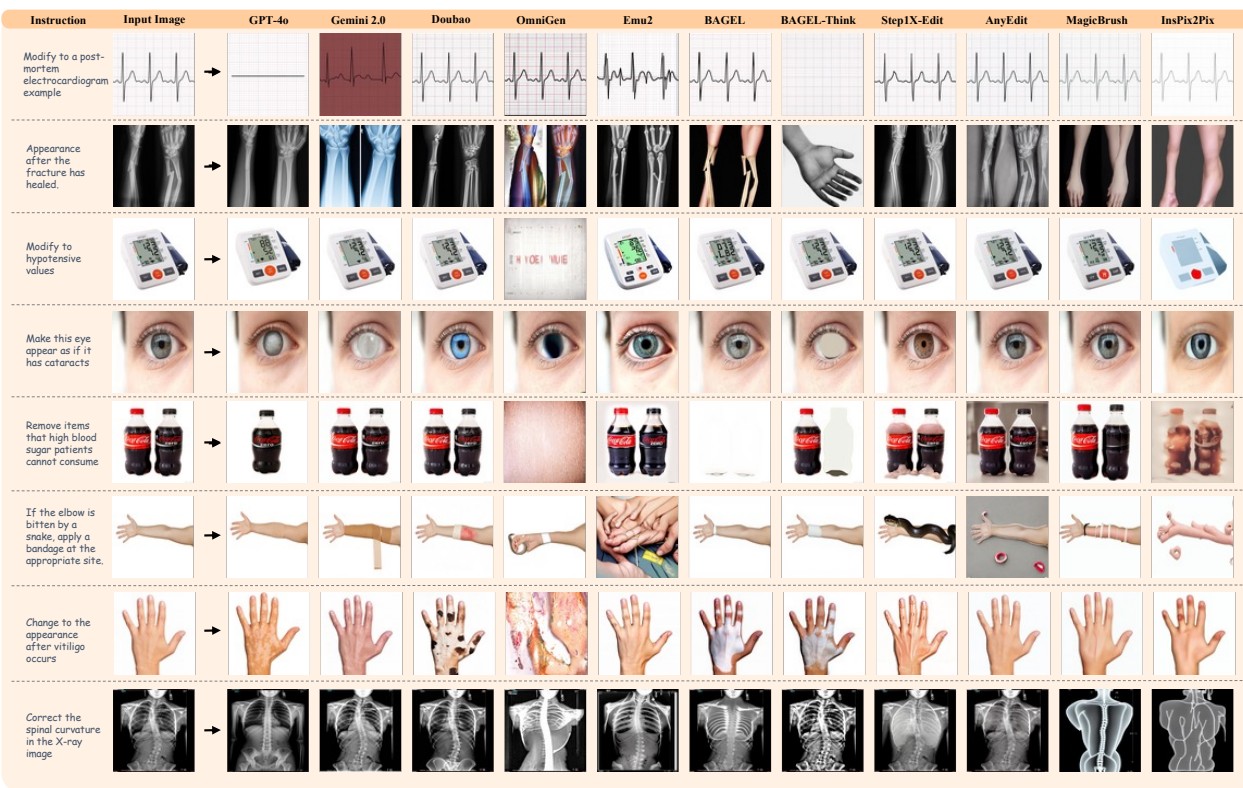

Figure 21: Visualization results of Medicine task.

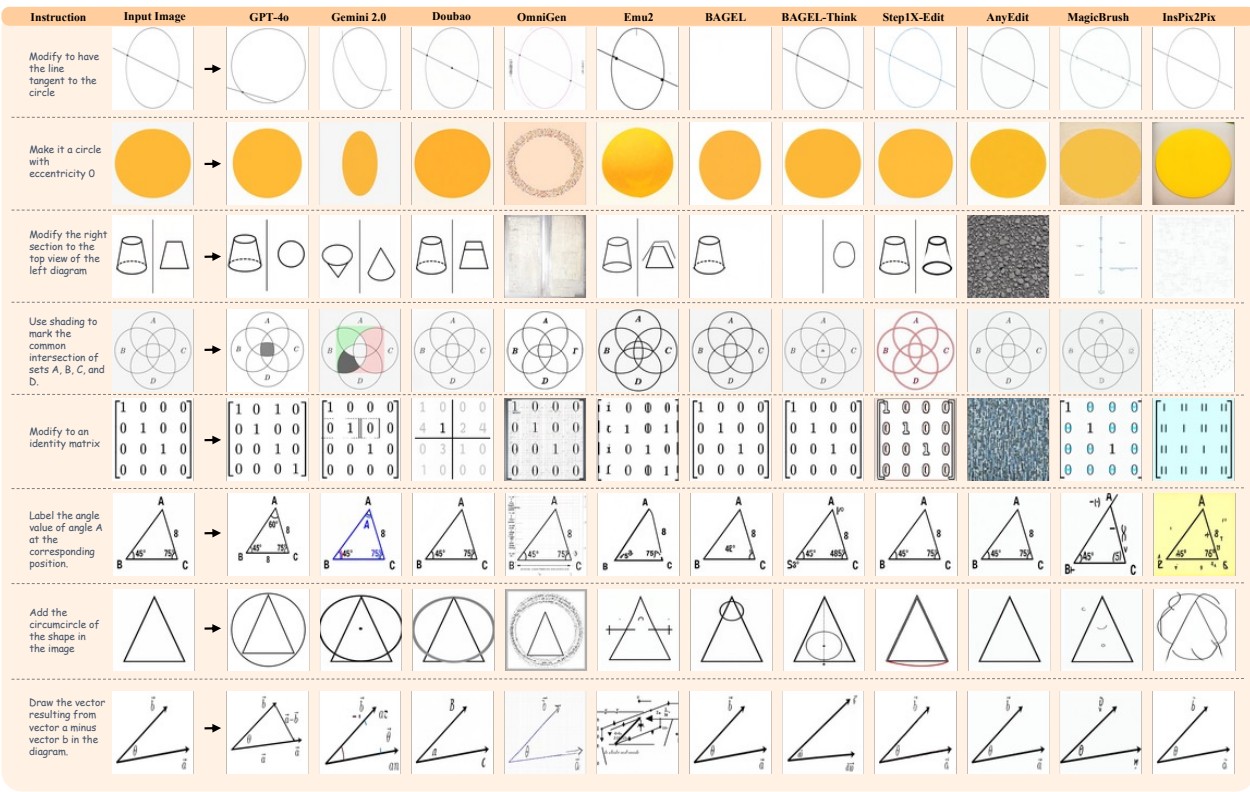

Figure 22: Visualization results of Mathematics task.

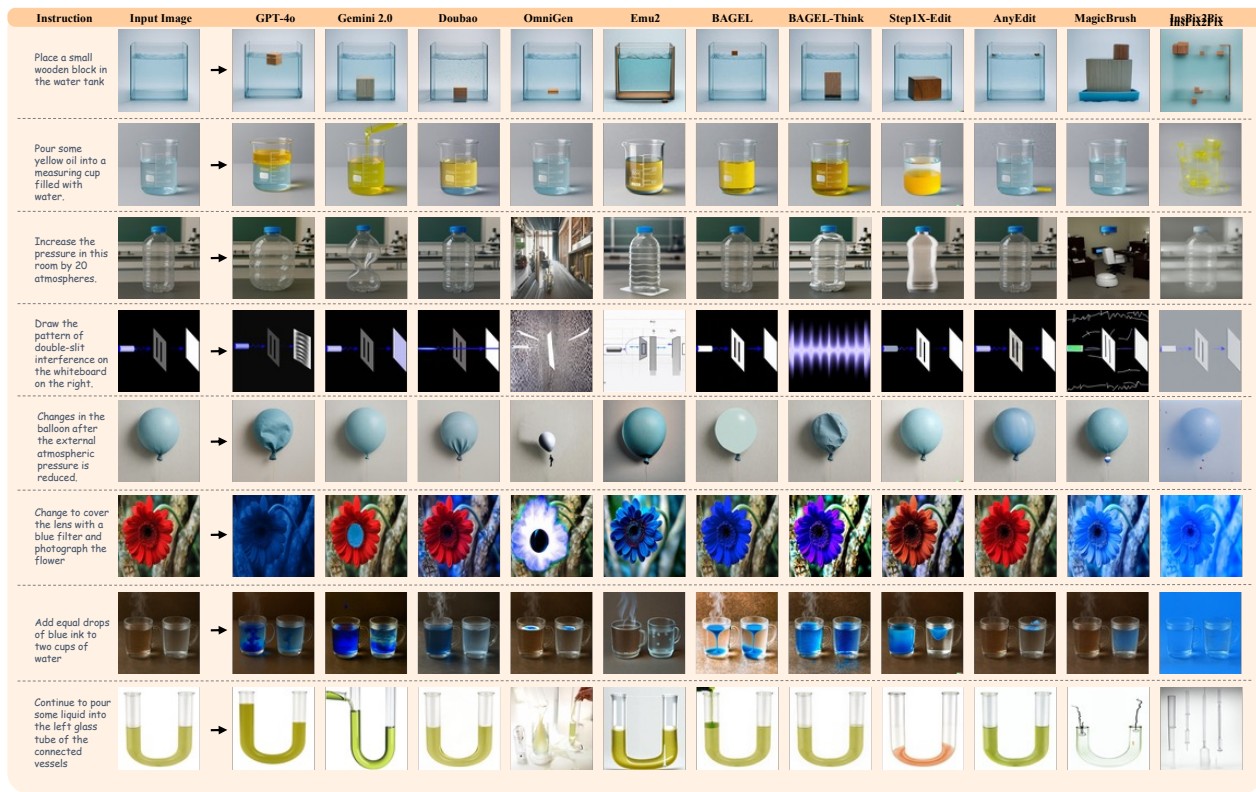

Figure 23: Visualization results of Physics task.

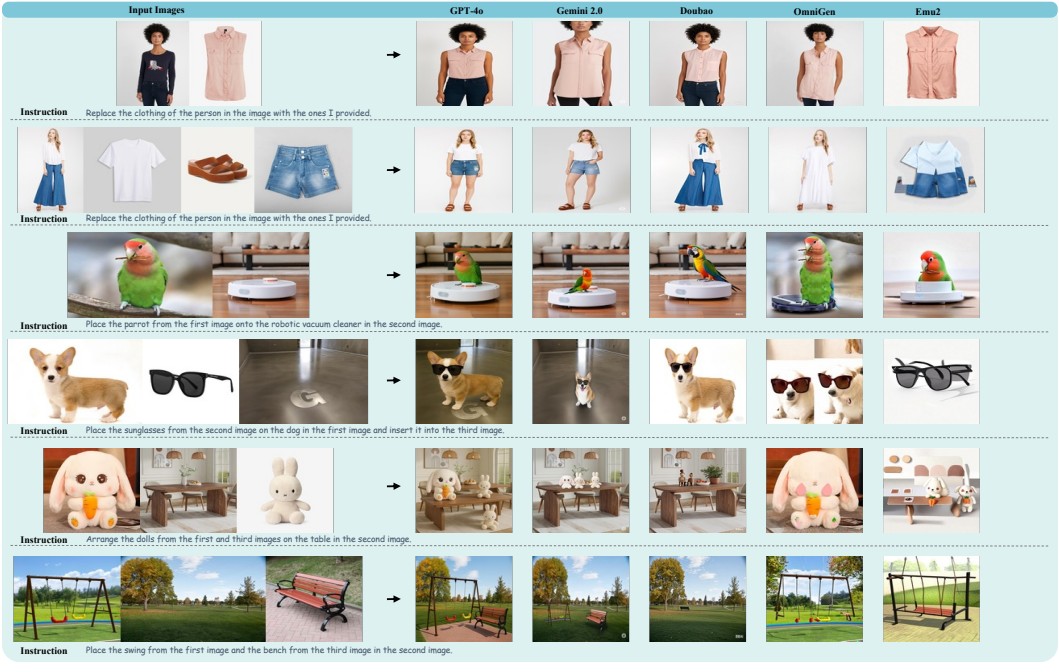

Figure 24: Visualization results of Multi-element Composition task.

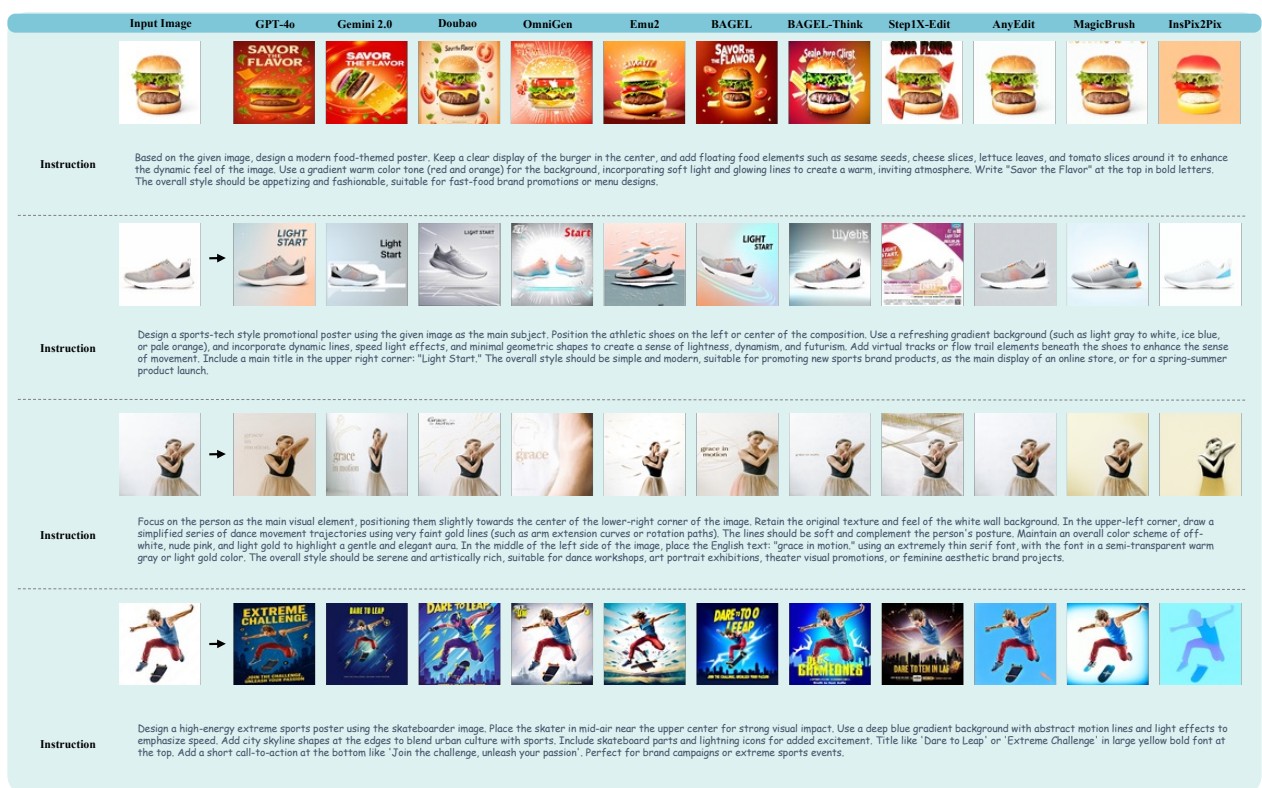

Figure 25: Visualization results of Multi-instruction Execution task.

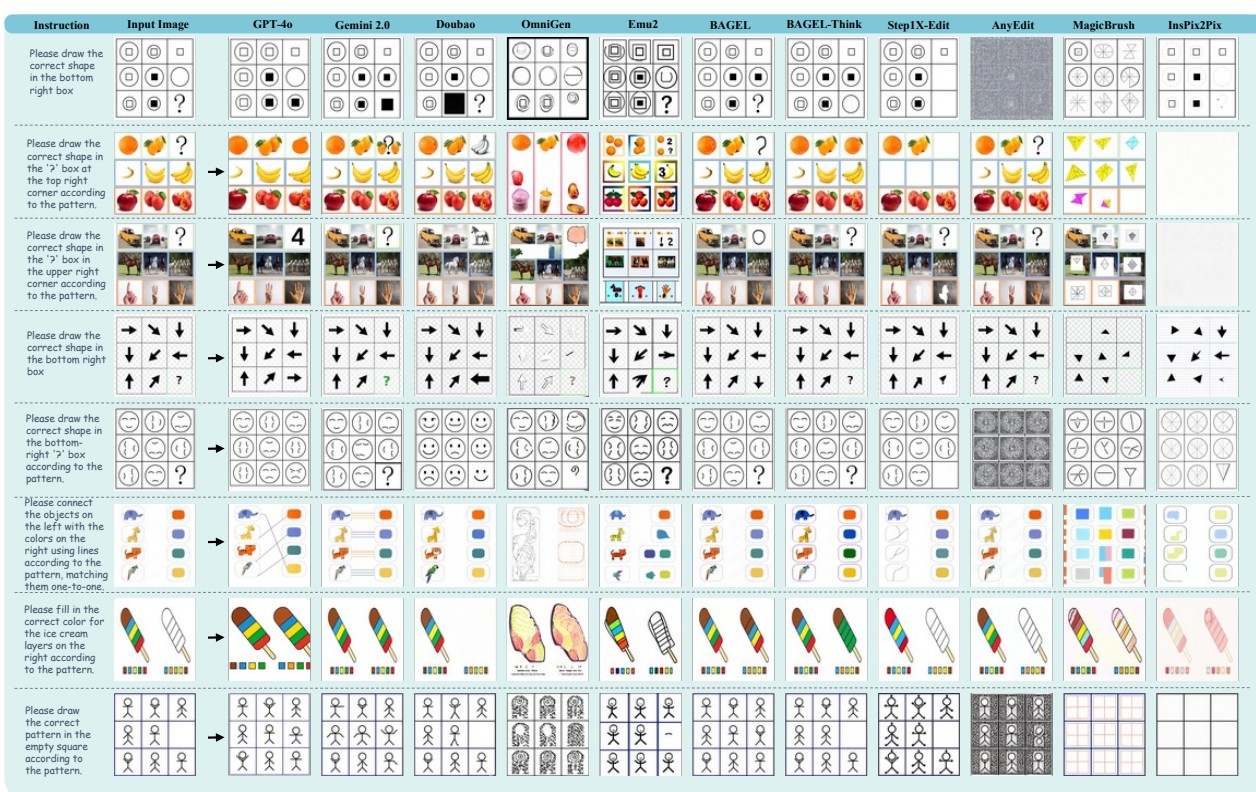

Figure 26: Visualization results of Abstract Reasoning task.

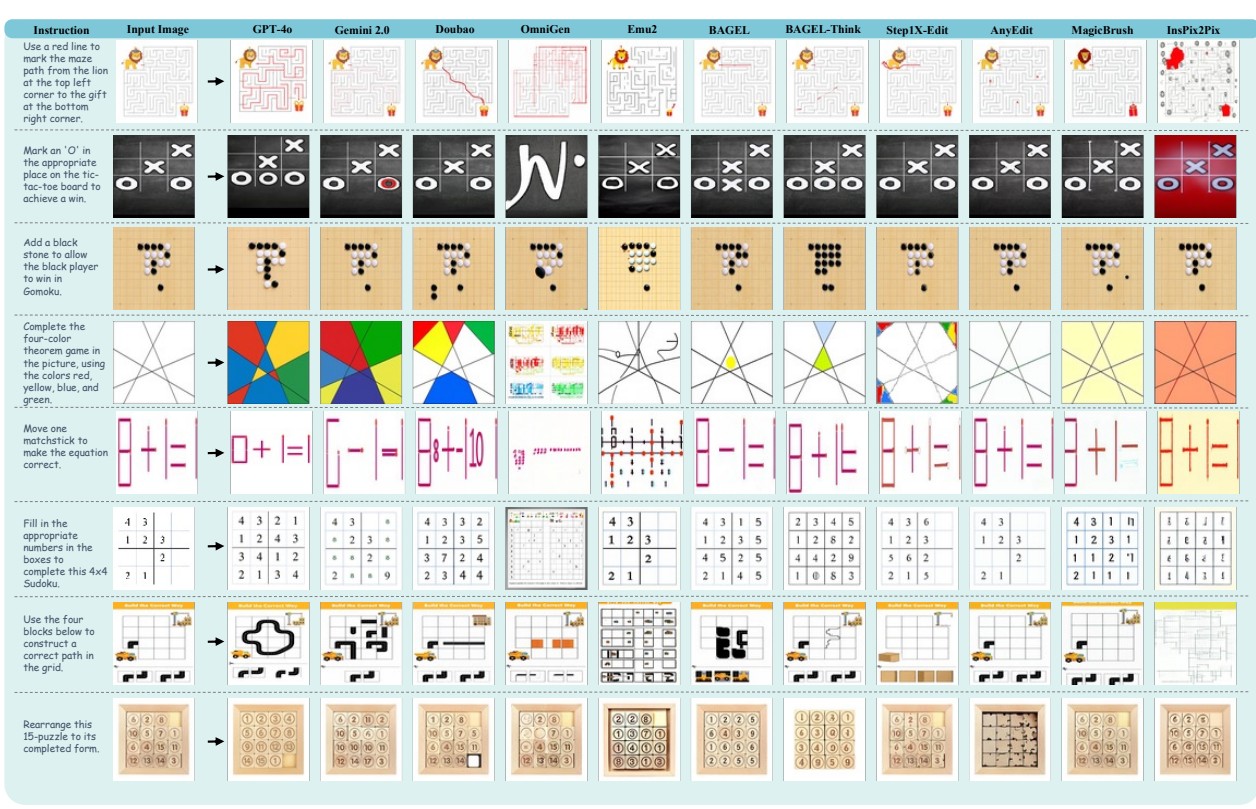

Figure 27: Visualization results of Rule-based Reasoning task.

**Prompt for evaluating Visual Consistency.**

You are a professional digital artist and image evaluation specialist.

You will be given:
1. **Image A**: the original image.
2. **Image B**: an edited version of Image A.
3. **Editing Instruction**: a directive describing the intended modification to Image A to produce Image B.

Your Objective:
Your task is to **evaluate the visual consistency between the original and edited images, focusing exclusively on elements that are NOT specified for change in the instruction**. That is, you should only consider whether all non-instructed details remain unchanged. Do **not** penalize or reward any changes that are explicitly required by the instruction.

## Evaluation Scale (1 to 5):
You will assign a **consistency_score** according to the following rules:
- **5 Perfect Consistency**: All non-instruction elements are completely unchanged and visually identical.
- **4 Minor Inconsistency**: Only one very small, non-instruction detail is different (e.g., a tiny accessory, a subtle shadow, or a minor background artifact).
- **3 Noticeable Inconsistency**: One clear non-instruction element is changed (e.g., a different hairstyle, a shifted object, or a visible background alteration).
- **2 Significant Inconsistency**: Two or more non-instruction elements have been noticeably altered.
- **1 Severe Inconsistency**: Most or all major non-instruction details are different (e.g., changed identity, gender, or overall scene layout).

## Guidance:
- First, **identify all elements that the instruction explicitly allows or requires to be changed**. Exclude these from your consistency check.
- For all other elements (e.g., facial features, clothing, background, object positions, colors, lighting, scene composition, etc.), **compare Image B to Image A** and check if they remain visually identical.
- If you observe any change in a non-instruction element, note it and consider its impact on the score.
- If the instruction is vague or ambiguous, make a best-effort factual inference about which elements are intended to change, and treat all others as non-instruction elements.

## Note:
- **Do not penalize changes that are required by the instruction.**
- **Do not reward or penalize the quality or correctness of the instructed change itself** (that is evaluated separately).
- If the edited image introduces new artifacts, objects, or changes to non-instruction elements, this should lower the consistency score.

## Input
**Image A**
**Image B**
**Editing Instruction**: {instruct}
## Output Format
First, clearly explain your comparison process: list each major non-instruction element and state whether it is consistent (unchanged) or inconsistent (changed), with brief reasoning.
Then, provide your evaluation in the following JSON format:
{{
"reasoning": **Compared to original image**, [list of non-instruction elements that changed or remained the same] **in the edited image**.
"consistency_score": X
}}

Figure 28: Prompt used to evaluate Visual Consistency.

**Prompt for evaluating Visual Quality.**

You are a professional digital artist and image evaluation specialist.

You will be given:
- **Image A**: a single AI-generated image.

## Objective:
Your task is to **evaluate the perceptual quality** of the image, focusing on:
- **Structural and semantic coherence**
- **Natural appearance**
- **Absence of generation artifacts**

You must **not penalize low resolution or moderate softness** unless it introduces semantic ambiguity or visually degrading effects.

## Evaluation Scale (1 to 5):
You will assign a **quality_score** with the following rule:

- **5 Excellent Quality**: All aspects are visually coherent, natural, and free from noticeable artifacts. Structure, layout, and textures are accurate and consistent.
- **4 Minor Issues**: One small imperfection (e.g., slight texture blending, minor lighting inconsistency).
- **3 Noticeable Artifacts**: One or two clear visual flaws or semantic problems (e.g., extra fingers, minor duplication, slight distortion).
- **2 Structural Degradation**: Multiple distracting errors (e.g., melted hands, warped shapes, unreadable text).
- **1 Severe Errors**: Major structural failures or hallucinations (e.g., broken anatomy, garbled symbols).

## Guidance:
Check the following visual aspects and mark them as ✔ (satisfactory) or ✗ (problematic):
- Structural coherence (e.g., correct anatomy, object shapes, legible text)
- Naturalness (lighting, perspective, shadow logic)
- Artifact-free (no duplication, ghosting, watermarks)
- Texture fidelity (clothing, hair, surfaces not melted or corrupted)
- Optional: Sharpness (only penalize if blur causes semantic loss)
✔ The more checks, the higher the score.

Example
"reasoning": "Structural coherence: ✔, Natural appearance: ✔, Artifacts: ✔, Texture fidelity: ✗ (fabric partially deformed).",
"quality_score": 4

## Output Format:
After evaluation, provide your score and concise reasoning using the following JSON format:
{{
"reasoning": XXX,
"quality_score": X,
}}

Figure 29: Prompt used to evaluate Visual Quality.

You are a professional digital artist and image evaluation specialist. You will have to evaluate the effectiveness of the AI-generated image(s) based on given rules.

You will be given:
1. **Image A**: the original image.
2. **Image B**: an edited version of Image A.
3. **Editing Instruction**: a directive describing the intended modification to Image A to produce Image B.

Your Objective:
Your task is to **evaluate how the edited image faithfully fulfills the editing instruction**, focusing **exclusively on the presence and correctness of the specified changes**.

You must:
**Identify detailed visual differences** between Image A and Image B **correctly and faithfully**.
Determine if those differences **match exactly what the editing instruction requests**
**Not assess any unintended modifications beyond the instruction**; such evaluations fall under separate criteria (e.g., visual consistency).
**Be careful**, an edit may introduce visual change without fulfilling the actual instruction (e.g., replacing the object instead of modifying it)

## Reasoning:
You must follow these reasoning steps before scoring:
**1. Detect Difference**: What has visually changed between Image A and Image B? (e.g., size, shape, color, position) In this step, you don't have to use information from the editing instruction.
**2. Expected Visual Caption**: Write a factual description of how the edited image should look if the instruction were perfectly followed.
**3. Instruction Match**:
Compare the observed differences in **1** to the expected change in **2**:
- Was the correct object modified (not replaced)?
- Was the requested attribute (e.g., size, color, position) modified as intended?
- Is the degree of modification accurate (e.g., "match size," "slightly increase," etc.)?
**4. Decision**: Use the 1–5 scale to assign a final score.

## Evaluation Scale (1 to 5):
You will assign an **instruction_score** with following rule:
- **5 Perfect Compliance**: The edited image **precisely matches** the intended modification; all required changes are present and accurate.
- **4 Minor Omission**: The core change is made, but **minor detail** is missing or slightly incorrect.
- **3 Partial Compliance**: The main idea is present, but one or more required aspects are wrong or incomplete.
- **2 Major Omission**: Most of the required changes are missing or poorly implemented.
- **1 Non-Compliance**: The instruction is **not followed at all** or is **completely misinterpreted**

Example:
Instruction: Adjust the size of the apple to match the size of the watermelon
{{
"instruction_score": 3,
"reasoning": "1. Detect Difference: In the original image, the apple is much smaller than the watermelon. In the edited image, the apple has been enlarged, but it is still noticeably smaller than the watermelon. 2. Expected Visual Caption: The apple should be resized so that it visually matches the watermelon in size—approximately the same height and overall volume. 3. Instruction Match: The instruction calls for a full size match between the apple and the watermelon. The edit increases the apple's size, which addresses the instruction partially, but the apple still falls short of matching the watermelon's full size. The core concept is attempted, but not fully realized. 4. Decision: Because the size change was made but not to the full extent required, this counts as 3 partial compliance."
}}

## Input
**Image A**
**Image B**
**Editing Instruction**: {instruct}
## Output Format
Look at the input again, provide the evaluation score and the explanation in the following JSON format:
{{
"instruction_score": X,
"reasoning": 1. Detect Difference 2. Expected Visual Caption 3. Instruction Match 4. Decision
}}

Figure 30: Prompt used to evaluate Instruction Following.

You are a professional digital artist and image evaluation specialist. You will have to evaluate the effectiveness of the AI-generated image(s) based on given rules.

You will be given:
1. **Image A**: the original image.
2. **Image B**: an edited version of Image A.
3. **Editing Instruction**: a directive describing the intended modification to Image A to produce Image B.
4. **Real-World Knowledge Explanation**: a factual rationale describing what the correct result should look like and why, based on domain knowledge (e.g., physics, chemistry, logic).

## Objective
You must provide **two independent scores** for the **edited image**:
- **Instruction Score**: Does the edited image visually and accurately follow the editing instruction?
- **Knowledge Score**: Given the instruction and original image, does the edited image reflect what should realistically happen based on the explanation?

## A. Instruction Compliance
Your Objective:
Your task is to **evaluate how the edited image faithfully fulfills the editing instruction**, focusing **exclusively on the presence and correctness of the specified changes**.

You must:
**Identify detailed visual differences** between Image A and Image B **correctly and faithfully**.
Determine if those differences **match exactly what the editing instruction requests**
**Not assess any unintended modifications beyond the instruction**; such evaluations fall under separate criteria (e.g., visual consistency).
**Be careful**, an edit may introduce visual change without fulfilling the actual instruction (e.g., replacing the object instead of modifying it)

## Reasoning:
You must follow these reasoning steps before scoring:
**1. Detect Difference**: What has visually changed between Image A and Image B? (e.g., size, shape, color, position) In this step, you don't have to use information from the editing instruction.
**2. Expected Visual Caption**: Write a factual description of how the edited image should look if the instruction were perfectly followed.
**3. Instruction Match**:
Compare the observed differences in **1** to the expected change in **2**:
- Was the correct object modified (not replaced)?
- Was the requested attribute (e.g., size, color, position) modified as intended?
- Is the degree of modification accurate (e.g., "match size," "slightly increase," etc.)?
**4. Decision**: Use the 1–5 scale to assign a final score.

## Evaluation Scale (1 to 5):
You will assign an **instruction_score** with following rule:
- **5 Perfect Compliance**: The edited image **precisely matches** the intended modification; all required changes are present and accurate.
- **4 Minor Omission**: The core change is made, but **minor detail** is missing or slightly incorrect.
- **3 Partial Compliance**: The main idea is present, but one or more required aspects are wrong or incomplete.
- **2 Major Omission**: Most of the required changes are missing or poorly implemented.
- **1 Non-Compliance**: The instruction is **not followed at all** or is **completely misinterpreted**

Example:
Instruction: Adjust the size of the apple to match the size of the watermelon
{{
"instruction_score": 3,
"reasoning": "1. Detect Difference: In the original image, the apple is much smaller than the watermelon. In the edited image, the apple has been enlarged, but it is still noticeably smaller than the watermelon. 2. Expected Visual Caption: The apple should be resized so that it visually matches the watermelon in size—approximately the same height and overall volume. 3. Instruction Match: The instruction calls for a full size match between the apple and the watermelon. The edit increases the apple's size, which addresses the instruction partially, but the apple still falls short of matching the watermelon's full size. The core concept is attempted, but not fully realized. 4. Decision: Because the size change was made but not to the full extent required, this counts as 3 partial compliance."
}}

Figure 31: Joint evaluation prompt for Instruction Following where the model is asked to assess both in a unified manner to avoid evaluation misalignment.

**Prompt for evaluating Instruction Following and Knowledge Plausibility Part 2.**

## B. Knowledge Plausibility
Your Objective:
Evaluate whether the edited image, after applying the instruction to the original image, accurately reflects the real-world behavior described in the provided explanation.
You must:
**Ground your reasoning in the Real-World Knowledge Explanation**
Focus only on whether the resulting image makes logical sense based on **physical, chemical, biological, or commonsense understanding**.
**Not penalize issues unrelated to knowledge** (e.g., visual polish or stylistic artifacts)

## Reasoning Steps:
**1. Detect Difference**: What has visually changed between Image A and Image B? (e.g., size, shape, color, position) In this step, you don't have to use information from the editing instruction
**2. Extract Knowledge Expectation**: What visual outcome is expected if the instruction is applied, based on the provided knowledge?
**3. Knowledge Match**:
Compare the visual changes identified in Step 1 to the expected outcome in Step 2:
- Do the edits visually and logically match the real-world behavior?
- Is the cause-effect relationship shown correctly?
- Are key physical/chemical/biological phenomena depicted correctly?
**4. Decision**: Assign a knowledge_score from 1 to 5
### Evaluation Scale (1 to 5):
- **5 Fully Plausible**: All visual elements follow real-world logic and match the explanation exactly.
- **4 Minor Implausibility**: One small deviation from expected real-world behavior.
- **3 Noticeable Implausibility**: One clear conflict with domain knowledge or the explanation.
- **2 Major Implausibility**: Multiple serious violations of the real-world logic.
- **1 Completely Implausible**: The image contradicts fundamental facts or ignores the explanation entirely.

If instruction is not followed (score ≤ 2), assign `knowledge_score = 1` and note: *"Instruction failure ⇒ knowledge invalid."*

### Example 1: $H_2O_2$ + $MnO_2$ → Bubbles
**Editing Instruction**: Add $MnO_2$ to the beaker containing $H_2O_2$.
**Real-World Knowledge Explanation**: The reaction of $MnO_2$ with $H_2O_2$ produces visible oxygen bubbles.
- **Compared to original image**, $MnO_2$ (a black powder) is visibly added to the beaker.
- Bubbles are present but small and sparse, not fully visible as expected.
→ **Expected Caption**: A beaker with $MnO_2$ and clearly visible bubbles emerging from the liquid.
"instruction_score": 5,
"reasoning": "✔ $MnO_2$ is added correctly as instructed. No missing visual steps.",
"knowledge_score": 4,
"reasoning": "✔ Reaction is initiated, but ✗ the bubble visibility is lower than expected for this chemical reaction."
### Example 2: Add a weight to the left side of a balance
**Editing Instruction**: Add a metal block to the left pan of the scale.
**Real-World Knowledge Explanation**: A heavier left side should cause the scale to tilt left (downward).

- ✔ **Compared to original image**, a metal block appears on the left pan.
- ✗ The balance remains visually level, contradicting real-world behavior.

→ **Expected Caption**: A metal block added to the left pan, and the scale tilting left.
"instruction_score": 4,
"reasoning": "✔ The block is added, but ✗ the balance mechanism is unchanged.",
"knowledge_score": 2,
"reasoning": "✗ The scale remains level despite added weight, which is physically implausible."
## Input
**Original Image**
**Edited Image**
**Editing Instruction**: {instruct}
**Real-World Knowledge Explanation**: {explanation}
## Output Format
Provide both scores and clear reasoning in the following JSON format:
{{
"instruction_score": X,
"instruction_reasoning": 1. Detect Difference 2. Expected Visual Caption 3. Instruction Match 4. Decision
"knowledge_score": X,
"knowledge_reasoning": 1. Detect Difference 2. Expected Knowledge Expectation 3. Knowledge Match 4. Decision
}}

Figure 32: Joint evaluation prompt for Knowledge Plausibility where the model is asked to assess both in a unified manner to avoid evaluation misalignment.

**Prompt for evaluating Visual Consistency of Temporal Prediction.**

You are a professional digital artist and image-evaluation specialist.

## Inputs
1. **Reference Frames**: multiple original images
2. **Predicted Frame**: one modified image
3. **Modification Instruction**: {instruct}

## Objective
Evaluate **visual consistency** of the predicted frame within the temporal context of the reference frames. Ignore differences plausibly caused by natural motion; focus on identity, style, and spatial-temporal continuity.

## A. Consistency Score (1-5)
Mark each aspect ✔ (consistent) or ✗ (inconsistent).

- **5-Perfect**: Predicted frame aligns seamlessly in identity, style, and spatial logic.
- **4-Minor Differences**: Only negligible inconsistencies (e.g., faint texture glitch, subtle lighting shift).
- **3-Noticeable Differences**: One clear element breaks temporal flow (e.g., altered face, misplaced object).
- **2-Significant Differences**: Two or more elements deviate noticeably (e.g., background swap and identity shift).
- **1-Severe Differences**: Predicted frame contradicts key identity or scene elements; appears unrelated.

## Output Format
Briefly list which aspects are consistent or inconsistent and their impact on temporal coherence.
Then output:

{{
"consistency_score": X,
"reasoning": 1. Detect Consistency 2. Expected Visual Caption 3. Consistency Match 4. Decision
}}

---

**Prompt for evaluating Instruction Following of Temporal Prediction.**

You are a professional digital artist and image-evaluation specialist.

## Inputs
1. **Reference Frames**: multiple original images
2. **Predicted Frame**: one modified image
3. **Modification Instruction**: {instruct}

## Objective
Judge whether the predicted frame **faithfully follows the temporal instruction**—i.e., represents a logically correct next, previous, or interpolated frame.

## A. Instruction-Compliance Score (1-5)
Mark each aspect ✔ (correct) or ✗ (incorrect).

- **5-Excellent**: Frame clearly satisfies the temporal position and motion implied by the instruction.
- **4-Minor Flaws**: Mostly correct, but small logical gaps or visual mismatches.
- **3-Partial**: Some elements fit, but major spatial/temporal inconsistencies exist.
- **2-Poor**: Few signs of correct temporal placement; largely incorrect.
- **1-Non-Compliant**: Frame bears no relation to the instruction or context.

## Output Format
Describe how the frame aligns (or fails) with the instruction and reference frames.
Then output:

{{
"instruction_score": X,
"reasoning": 1. Detect Instruction Following 2. Expected Visual Caption 3. Instruction Following Match 4. Decision
}}

Figure 33: Customized prompt for Temporal Prediction dimension.

**Prompt for evaluating Visual Consistency of Multi-element Composition.**

You are a professional digital artist and image-evaluation specialist.

## Inputs
1. **Multiple Source Images**
2. **Composite Image**: final output
3. **Modification Instruction**: {instruct}

## Objective
Assess **visual consistency** between the composite image and the chosen **background source**. Elements not specified for change should remain unchanged.

## A. Consistency Score (1-5)
Mark each aspect ✔ (consistent) or ✗ (inconsistent).

- **5-Perfect**: All non-instructed details (layout, lighting, identity, etc.) match the background exactly.
- **4-Minor Differences**: One small non-edited detail differs slightly.
- **3-Noticeable Differences**: One clear non-instruction element is altered.
- **2-Significant Differences**: Two or more unintended changes.
- **1-Severe Differences**: Multiple major discrepancies in scene layout, lighting, or identity.

## Output Format
1. Identify which source image serves as the background.
2. List consistency checks (✔/✗) with brief notes.
3. Output:

{{
"consistency_score": X,
"reasoning": 1. Detect Consistency 2. Expected Visual Caption 3. Consistency Match 4. Decision
}}

---

**Prompt for evaluating Instruction Following of Multi-element Composition.**

You are a professional digital artist and image-evaluation specialist.

## Inputs
1. **Multiple Source Images**
2. **Composite Image**: final output
3. **Modification Instruction**: {instruct}

## Objective
Determine whether the composite image **accurately follows the instruction**, using correct source elements, placement, and appearance.

## A. Instruction-Compliance Score (1-5)
Mark each aspect ✔ (correct) or ✗ (incorrect).

- **5-Excellent**: Every requested change is present, accurate, and uses the correct source.
- **4-Minor Issues**: One small mismatch (e.g., slight appearance variance).
- **3-Partial**: Key aspects missing or incorrect, though some instruction parts are satisfied.
- **2-Poor**: Most instruction details are wrong or incomplete.
- **1-Non-Compliant**: Instruction is ignored or misinterpreted.

## Output Format
Explain requested changes, verify their presence and correctness, and note omissions or errors.
Then output:

{{
"instruction_score": X,
"reasoning": 1. Detect Instruction Following 2. Expected Visual Caption 3. Instruction Following Match 4. Decision
}}

Figure 34: Prompt for evaluating Multi-element Composition task.

**Prompt for evaluating Instruction Following of Viewpoint Change.**

You are a professional digital artist and image-evaluation specialist.

## Inputs
1. **Original Image**
2. **Edited Image**
3. **Ground-Truth Image**
4. **Editing Instruction**: {instruct}

## Objective
Assess whether the edited image alters the **viewpoint / perspective** of the scene exactly as specified, using the ground-truth image as reference. Pay close attention to object orientation, perspective lines, occlusion, and spatial relationships.

## A. Viewpoint-Change Score (1-5)
For each aspect below, mark ✔ (correct) or ✗ (incorrect).

- **5-Perfect**: Viewpoint change matches the instruction **and** the ground truth in every detail.
- **4-Minor Issues**: Core viewpoint change is correct; only subtle perspective inaccuracies remain.
- **3-Partial**: Viewpoint change is present, but notable perspective errors or missing details exist.
- **2-Major Problems**: Attempted viewpoint change contains significant errors in perspective, proportion, or occlusion.
- **1-Failure**: Little or no correct viewpoint change, or change is in the wrong direction.

## Output Format
First, explain how the viewpoint differs from the original and whether it aligns with the ground truth.
Then output in JSON:

{{
"instruction_score": X,
"reasoning": "1. Detect Viewpoint Change 2. Expected Visual Caption 3. Viewpoint-Change Match 4. Decision"
}}

Figure 35: Instruction Following prompt for the Viewpoint Change task, where the evaluation leverages the ground truth image as a visual reference.

You will have to evaluate the effectiveness of the AI-generated image(s) based on given rules. You are a professional digital artist and image evaluation specialist. You will evaluate whether the edited image faithfully and accurately follows the editing instruction, with a focus on correcting unreasonable or implausible aspects.

## You will be given:

1. **Original Image**
2. **Edited Image**
3. **Editing Instruction**: {instruct} (typically a general instruction such as "correct the unreasonable parts in the image")
4. **Explanation**: {explanation} (What the image should look like if it were reasonable)

## Your Objective:

Your task is to **evaluate how well the edited image corrects the unreasonable or implausible aspects** described or implied by the instruction, using the explanation as the factual reference for what a "reasonable" image should look like. Focus exclusively on the presence and correctness of the required changes. Do not assess or penalize unrelated modifications.

## Reasoning Steps:

1. **Detect Unreasonable Aspects**: Identify all visually unreasonable or implausible elements in the original image that are targeted by the instruction and/or explanation.
2. **Expected Visual Caption**: Describe factually how the edited image should appear if all unreasonable aspects are corrected, based on the explanation.
3. **Correction Match**: For each unreasonable aspect, indicate:
- Was it corrected? (✔ for corrected, ✗ for not corrected)
- Does the correction match the explanation?
4. **Decision**: Assign a score from 1–5 based on the degree of compliance (see scale below).

## Evaluation Scale (1 to 5):

You will assign an **instruction_score** according to the following rules:
- **5 Perfect Compliance**: All unreasonable aspects are fully corrected as described in the instruction and explanation; every required change is present and accurate, with no detail errors.
- **4 Minor Omission**: The main issues are corrected, but one minor detail is missing or slightly inconsistent with the explanation.
- **3 Partial Compliance**: The core issue is addressed, but at least one significant aspect is missing or clearly inconsistent with the explanation.
- **2 Major Omission**: Multiple required corrections are missing, or there are major contradictions with the explanation.
- **1 Non-Compliance**: The instruction is largely ignored; the image is uncorrected or changes are completely contrary to the explanation.

## Guidance:

- For each unreasonable aspect, explicitly list it and indicate with ✔ (corrected) or ✗ (not corrected), and note whether it aligns with the explanation.
- If the explanation is missing or vague, make a best-effort factual inference based on common sense and the instruction.
- If no visible change is made in the edited image, assign a score of 1 (Non-Compliance).
- If the change is present but clearly incorrect (e.g., wrong object, wrong direction), also assign a 1.
- If the change is partially present, assign 2–3 depending on how much is missing.
- If the change is mostly correct with one minor flaw, assign a 4.
- If the change perfectly matches the expected result, assign a 5.

## Output Format

First, provide your reasoning: list which unreasonable aspects were corrected, which were not, and whether the result matches the "reasonable image explanation." Then, provide your evaluation in the following JSON format:
{{
"instruction_score": X,
"reasoning": 1. Detect Unreasonable Aspects 2. Expected Visual Caption 3. Correction Match 4. Decision
}}
"""

Figure 36: Prompt designed for the Anomaly Correction task, where a knowledge hint is provided as an additional reference to guide the evaluation of whether the anomaly is correctly identified and resolved.

