# OpenReview forum: "KRIS-Bench: Benchmarking Next-Level Intelligent Image Editing Models"
_NeurIPS.cc/2025/Datasets_and_Benchmarks_Track — NeurIPS 2025 Datasets and Benchmarks Track poster_

### Official Review · Reviewer_68Wp · 2025-06-29

**Rating:** 5
**Confidence:** 4

**Summary:**

This paper proposes a knowledge-based image editing benchmark. This work is inspired by education theory and includes three types of knowledge: Factual, Conceptual, and Procedural. It covers 22 representative tasks spanning 7 reasoning dimensions and releases 1,267 high-quality annotated editing instances. For evaluation, it introduces four criteria, including Visual Consistency, Visual Quality, Instruction Follow, and Knowledge Plausibility. Extensive experiments are conducted on both closed-source and open-source models, and the results show some insightful analysis. Overall, I think it is a great work for the community.

**Additional Feedback:**

1. For dataset construction, human annotators are adopted. How could we ensure the quality of the dataset?
2. As shown in the limitation, is there any way to enlarge the evaluation dataset?
3. As for knowledge type, is there any other knowledge type beyond this education theory that could be adopted?
4. Are there any methods (such as COT for LLM) to improve the generation performance for all methods?

**Dataset Code Accessibility:**

Yes

**Dataset Code Comments:**

Both the code and datasets could be downloaded.

**Ethical Considerations:**

No, there are no or only very minor ethics concerns

**Final Justification:**

5

**Limitations Weaknesses:**

1. For dataset construction, human annotators are adopted. How could we ensure the quality of the dataset?
2. As shown in the limitation, is there any way to enlarge the evaluation dataset?
3. As for knowledge type, is there any other knowledge type beyond this education theory that could be adopted?
4. Are there any methods (such as COT for LLM) to improve the generation performance for all methods?

**Strengths Contributions:**

1. This paper focuses on an important problem where knowledge is a fundamental requirement for the image editing task. As we usually need the generated image to fit the physical world.  I think the motivation is good and worthy of exploration.
2. Based on education theory,  it covers three types of knowledge, Factual, Conceptual, and Procedural, which are most common in our lives.
3. It includes four criteria, and it could both evaluate the image quality and the knowledge needed for this scenario. I think it is a good evaluation design.
4. Extensive experiments are conducted, and several analyses are shown.
5. A human study is conducted to show the effectiveness of VLM-as-a-judge.

---

> ### Author Rebuttal · Authors · 2025-07-31
>
> We sincerely appreciate your constructive suggestions for improving our paper. We are grateful for your recognition of our taxonomy grounded in educational theory, as well as the design of the four evaluation criteria. We're also pleased that you found the experiments and human study to be valuable.
>
> We hope our responses adequately address your concerns:
>
> **1. Dataset Quality Assurance**
>
> To ensure high-quality annotations, editing instructions were initially drafted by trained annotators and then refined using ChatGPT to enhance linguistic diversity and realism under human supervision. All annotations underwent a thorough review by a panel of experts holding Ph.D. degrees. For tasks requiring domain-specific knowledge, we further engaged reviewers with relevant expertise. This layered process strikes a balance between scalability and rigorous quality control.
>
> **2. Evaluation Set Expansion**
>
> While the current evaluation set is limited in size due to the manual effort involved, it is still larger than comparable benchmarks such as RISEBench [1] (360 samples). We plan to expand it through two complementary approaches: (1) semi-automatic augmentation by generating candidate samples via models followed by human validation, and (2) community-driven contributions through expert crowdsourcing or benchmark competitions. These strategies can help scale the evaluation set more efficiently.
>
> **3. Broader Knowledge Types**
>
> Alternative frameworks, such as the SECI model of knowledge dimensions [2], are indeed applicable in principle. However, we found that educational taxonomies (e.g., factual, conceptual, procedural knowledge) align more closely with the observable reasoning processes in image editing tasks and are easier to infer from the editing context. We therefore adopted this taxonomy for its greater practical relevance in our setting.
>
> **4. Improving Generation Performance**
>
> We agree that incorporating explicit reasoning steps can improve generation quality. Recent work such as Bagel [3] shows that introducing a structured "thinking" process (Chain-of-Thought), significantly enhances performance on multimodal tasks.
>
> Below are two output examples of reasoning in biology and chemistry tasks from Bagel:
>
>
>
> 1. The user wants to demonstrate the effect of planting hydrangeas in acidic soil on their color. **The input image shows hydrangeas with mixed blue and purple hues, which are influenced by soil pH. The output image should depict a clear color shift—hydrangeas turning predominantly pink or red—reflecting the impact of acidic conditions.** The structure and layout should remain consistent, with only the flower color changed.
>
>
>
>
> 2. The task involves adding barium chloride to a sulfuric acid solution. **This reaction produces barium sulfate, a white precipitate. The answer image should visually show the addition of barium chloride and the formation of the white precipitate.** The original bottle and liquid should remain unchanged, except for the visible appearance of the precipitate in the solution.
>
>
> These examples illustrate how explicitly modeling the reasoning process via Chain-of-Thought enables the system to correctly infer and generate the desired image edits.
>
> [1] Envisioning Beyond the Pixels: Benchmarking Reasoning-Informed Visual Editing
>
> [2] Management of knowledge creation
>
> [3] Emerging Properties in Unified Multimodal Pretraining

---

> > ### Comment · Reviewer_68Wp · 2025-08-03
> > **Response**
> >
> > Thanks for your response. After reading the rebuttal, most of my concerns are addressed and I will keep my score.

---

> > > ### Author Response · Authors · 2025-08-03
> > >
> > > Thank you very much for taking the time to review our paper and for your thoughtful comments. We appreciate your engagement with our rebuttal and are glad to hear that your concerns have been addressed.

---

### Official Review · Reviewer_hCAc · 2025-06-30

**Rating:** 4
**Confidence:** 4

**Summary:**

1. This paper introduces a knowledge-aware image editing task, outlining the necessary knowledge requirements for such edits.

2. The authors benchmark several existing image editing models and employ ChatGPT/LLMs as automated scoring models for evaluation.

3. A user study is conducted to compare human judgments with LLM-based scoring.

**Dataset Code Accessibility:**

Yes

**Ethical Considerations:**

No, there are no or only very minor ethics concerns

**Final Justification:**

The authors addressed most of my concerns, thus I am increasing my ratings.

**Limitations Weaknesses:**

1. Reliability and generalization of LLM-based Scoring: Relying solely on black-box models (e.g., ChatGPT or VLMs) may introduce hallucinations, limiting reproducibility. I suggest the authors to combine rule-based metrics (e.g., object detection for counting, temporal consistency checks) with model-based scoring for robustness.

2. Depth of User Study: While high correlation between LLM and human ratings is noted, a detailed breakdown of performance gaps is missing. As a benchmark paper, it should provide deeper analysis on why and how LLMs succeed/fail in scoring edits, rather than just reporting agreement rates.

**Strengths Contributions:**

1. Relevance & Contribution: Knowledge-aware image editing is a valuable and emerging research direction. Establishing a standardized benchmark would greatly benefit the community.

2. Evaluation Approach: Leveraging LLMs for automated scoring is an interesting and efficient way to assess model performance at scale.

---

> ### Author Rebuttal · Authors · 2025-07-31
>
> We sincerely appreciate you for constructive suggestions to improve the paper. We hope our response can address your concerns:
>
> **1. Reliability and Generalization of VLM-based Scoring**
>
> We appreciate your concern regarding the limitations of using vision-language models (VLMs) as evaluators. To mitigate hallucination risks and ensure generalization and reproducibility, we do not treat VLM evaluators as opaque black boxes. Instead, we design task-specific prompts, incorporate knowledge hints, and adopt multi-dimensional scoring to guide the model toward context-aware and grounded evaluations. Crucially, our prompts are crafted to elicit intermediate reasoning steps, enabling inspection of the model’s decision-making process and enhancing interpretability.
>
> To ensure reproducibility and model-agnostic consistency, we further include evaluations using the open-source model Qwen2.5-VL-72B, whose task-level trends closely align with those of GPT-4o (see Table 3 and Figure 7), demonstrating the robustness of our protocol.
>
> We also draw support from several recent studies: [1] demonstrates GPT-4V's strong alignment with human judgments across I2T, T2I, and I2I tasks, especially in semantic consistency and perceptual realism for text-guided image editing (Section 3.3); [2] shows that GPT-4's evaluations of text-image alignment outperform traditional automated metrics like CLIPScore in human alignment; [3] indicates that OpenAI's recent models provide evaluation results highly aligned with human judgments on visual quality tasks. And the user study involving 12 domain experts conducted in our work reveals strong correlations between human judgments and VLM-generated scores.
>
> To assess whether VLMs can handle explicit rule-based tasks within our benchmark, we conduct an experiment on the Count Change task in KRIS-Bench. The results show that GPT-4o achieves 96% accuracy, suggesting that even low-level attribute assessments can be reliably handled by VLMs.
>
> Besides, many of the evaluation aspects in KRIS-Bench, such as assessing whether the model has applied the correct scientific knowledge, followed complex instructions, or generated plausible multi-element edits, go beyond what current rule-based tools (e.g., object detectors) can reliably handle. As our benchmark focuses on high-level reasoning and knowledge alignment, we believe that VLM-based judgment provides a more scalable and cognitively aligned evaluation method than purely rule-based heuristics.
>
> While our current VLM-based evaluation shows good alignment with human reasoning and generalizes across models, we agree that hybrid evaluation paradigms could further enhance robustness. We will include this discussion in the revised version to highlight promising directions for integrating symbolic and model-based evaluators in future work.
>
> [1] GPT-4V(ision) as a Generalist Evaluator for Vision-Language Tasks
>
> [2] Exploring GPT-4 Vision for Text-to-Image Synthesis Evaluation
>
> [3] Comparative Evaluation of Multimodal Large Language Models for No-Reference Image Quality Assessment
>
> **2. Depth of User Study**
>
> Thank you for the suggestion. We have prepared a more comprehensive summary table presenting the user study results for each model across the four evaluation metrics. This allows us to more clearly observe where LLM-based evaluations align with or diverge from human judgments:
>
>
> | Metric                 | Model   | Pearson r | MAE    |
> | ---------------------- | ------- | --------- | ------ |
> | Visual Consistency     | GPT4o     | 0.4649    | 0.8130 |
> |                        | Gemini 2.0| 0.8053    | 0.4463 |
> |                        | Doubao| 0.9214    | 0.3141 |
> | Visual Quality         | GPT4o| 0.7097    | 0.1915 |
> |                        | Gemini 2.0| 0.6170    | 0.2740 |
> |                        | Doubao| 0.6602    | 0.2821 |
> | Instruction Following  | GPT4o     | 0.6514    | 0.5227 |
> |                        | Gemini 2.0| 0.6065    | 0.6301 |
> |                        | Doubao  | 0.7479    | 0.6287 |
> | Knowledge Plausibility | GPT4o     | 0.9467    | 0.6815 |
> |                        | Gemini 2.0| 0.9343    | 0.5559 |
> |                        | Doubao| 0.9587    | 0.4802 |
>
> As shown in the table, the correlation between human ratings and VLM-based scores for Visual Consistency is high for Gemini 2.0 (0.8053) and Doubao (0.9214), but noticeably lower for GPT-4o (0.4649). We attribute this to GPT-4o’s tendency to subtly enhance image details or alter the overall style and color tone, changes that may go unnoticed by VLMs but are often deemed important by human evaluators.
>
> In contrast, for Visual Quality, Instruction Following, and Knowledge Plausibility, all three models exhibit stronger correlations (from 0.6065 to 0.9587) between VLM and human ratings, demonstrating the reliability of VLM-based evaluation on these aspects.
>
> In addition, we present the correlation between the evaluation results from Qwen2.5-VL-72B and the user study as follows:
>  | Metric          | Model  | Pearson r | MAE    |
> | -------------------------- | ------ | --------- | ------ |
> | Visual Consistency     | GPT4o    | 0.5626    | 1.2847 |
> |                            | Gemini 2.0 | 0.7693    | 0.9678 |
> |                            | Doubao | 0.9014    | 0.5865 |
> | Visual Quality         | GPT4o   | 0.6138    | 0.2226 |
> |                            |  Gemini 2.0| 0.6248    | 0.3179 |
> |                            | Doubao | 0.6184    | 0.2877 |
> | Instruction Following  | GPT4o    | 0.5346    | 0.6854 |
> |                            |  Gemini 2.0| 0.6481    | 0.7702 |
> |                            | Doubao | 0.6731    | 0.7862 |
> | Knowledge Plausibility | GPT4o    | 0.8784    | 0.6069 |
> |                            |  Gemini 2.0| 0.9063    | 0.6360 |
> |                            | Doubao | 0.9147    | 0.5854 |
>
> The results show that, in most cases, the scoring by Qwen2.5-VL-72B remains highly correlated with human judgments. Notably, for the Visual Consistency of GPT-4o, Qwen2.5-VL-72B demonstrates a higher correlation (0.5626) with human evaluations than GPT-4o's own scores. This may be attributed to a potential bias of GPT-4o in favor of its own generated outputs. We will further elaborate on this issue in the revised version.
>
> We also conducted a detailed analysis of the failure cases in VLM scoring. We found that a significant portion of the errors fall into the Size Adjustment category, likely due to the VLM's insensitivity to relative spatial sizes between objects. Another subset of failures stems from particularly complex reasoning, such as rule-based logic, where the VLM exhibits instability when scoring highly difficult problems. A more fine-grained analysis will be provided in the revision.
>
>
> Although the use of LLM-as-a-judge is not the core claim and contribution of our paper, we agree that a deeper analysis is valuable. In the revision, we will include a detailed discussion of where and why LLM-based scoring succeeds or fails. This analysis will help the community better understand the current limitations of automated evaluation and inform future improvements in benchmark design.

---

> > ### Comment · Reviewer_hCAc · 2025-08-05
> >
> > The authors addressed most of my concerns, I will be increasing my ratings accordingly.

---

> > > ### Author Response · Authors · 2025-08-05
> > >
> > > Thank you very much for taking the time to review our paper. We are glad to hear that your concerns have been addressed.

---

### Official Review · Reviewer_tbw8 · 2025-06-30

**Rating:** 6
**Confidence:** 4

**Summary:**

This paper introduces KRIS-Bench, a new benchmark designed to evaluate instruction-based image editing models through the lens of knowledge-based reasoning. Inspired by educational theory, the authors classify editing tasks according to three types of knowledge, Factual, Conceptual, and Procedural, and define seven reasoning dimensions spanning 22 distinct task types. The dataset includes 1,267 annotated image editing instances, each accompanied by detailed instructions and, where appropriate, knowledge hints to aid evaluation. The authors also introduce a novel evaluation dimension, Knowledge Plausibility, which assesses whether a model's output is consistent with real-world knowledge. This is operationalized through structured prompts and validated via human studies. Experiments across 10 open- and closed-source models demonstrate substantial gaps in current models’ ability to reason under knowledge-intensive instructions.

**Additional Feedback:**

Although the authors provide extensive visual results in the supplementary material, the findings could be made more insightful and practically valuable by including model-level qualitative failure analyses. This would offer deeper understanding of common failure modes and provide clear guidance for future model improvements.

**Dataset Code Accessibility:**

Yes

**Ethical Considerations:**

No, there are no or only very minor ethics concerns

**Final Justification:**

After reading the reviews and rebuttal, I believe this work makes a strong and timely contribution. I agree with Reviewer 68Wp on the benchmark’s clear motivation and sound evaluation. Thus, I will raise my score.

**Limitations Weaknesses:**

1. **Single-turn setting:** The benchmark exclusively targets single-turn image editing. Many real-world editing scenarios involve iterative, multi-turn refinement, which this benchmark does not currently support.
2. **Model compatibility gaps:** Some tasks require multi-image input, which several open-source models cannot process. As a result, they receive a score of zero on these tasks, skewing the overall evaluation.
3. **Lack of clarity in Table 2 scoring:** The method for calculating scores in Table 2 is not clearly explained, particularly the "average score." It is unclear how the averages are computed across different metrics and dimensions. Here, we assume that the average is taken over each metric and dimension.
4. **Font readability issues:** Some font sizes are too small to be legible, for example, in Figure 4.

**Strengths Contributions:**

1. **Novel framing:** The use of a cognitively grounded taxonomy provides a principled and interpretable way to dissect reasoning demands in image editing tasks.
2. **Comprehensive task design:** The benchmark spans 22 tasks across 7 reasoning categories, covering diverse domains including physics, biology, social commonsense, mathematics, and symbolic reasoning. This results in broad and nuanced coverage of reasoning types rarely evaluated in current benchmarks.
3. **New evaluation metric:** The Knowledge Plausibility score is an important and underexplored addition. The combination of knowledge hints and model evaluation using tailored VLM prompts enables more robust and fine-grained assessments of model reasoning.

---

> ### Author Rebuttal · Authors · 2025-07-31
>
> Thank you for your positive comments. We're glad you found the cognitive framing useful and the task coverage broad and meaningful. We also appreciate your support for the Knowledge Plausibility score and the use of knowledge hints in evaluation, it’s great to know you see value in this direction. Below we address the four main limitations raised:
>
> **1. Single-turn Editing Only**
>
> We acknowledge that many real-world editing scenarios require multi-turn, iterative interaction. However, our goal in this work is to establish a principled foundation for studying knowledge-based reasoning in instruction-based image editing. To this end, we intentionally focus on the single-turn setting, which allows for better control of variables and clearer attribution of reasoning behavior. We view multi-turn extensions as an important direction for future work and plan to explore it in follow-up benchmarks.
>
> **2. Model Compatibility with Multi-image Input**
>
> We appreciate the concern regarding models that fail to process tasks requiring multi-image inputs. In the revision, we will include an additional leaderboard that only considers single-image tasks, to allow fairer comparisons among models with limited input capabilities. That said, we believe the ability to handle multi-image inputs is an important aspect of a model's practical utility, even though it is not strictly part of knowledge reasoning. Therefore, we choose to retain multi-image tasks in the overall leaderboard.
>
> **3. Score Averaging in Table 2**
>
> We thank the reviewers for pointing out the lack of clarity. The average scores in Table 2 are computed as weighted averages based on the number of examples per task. We will make this explanation explicit in the revision to avoid any ambiguity regarding how metrics are aggregated across different dimensions.
>
> **4. Font Size and Figure Readability**
>
> We appreciate the comment on figure readability. We will revise the layout and increase font sizes in Figure 4 to ensure they are legible.

---

> > ### Comment · Reviewer_tbw8 · 2025-08-03
> >
> > After carefully reading other reviews and the authors' rebuttal, I believe this work makes a strong and timely contribution. I agree with Reviewer 68Wp on the benchmark’s clear motivation, good evaluation designs. While some limitations remain, I find the authors’ responses reasonable and their roadmap for future extensions convincing. Overall, I think this paper will be valuable to the community and recommend acceptance.

---

> > > ### Author Response · Authors · 2025-08-03
> > >
> > > Thank you for your kind recognition of our work, we truly appreciate it.

---

### Official Review · Reviewer_4uTV · 2025-07-04

**Rating:** 4
**Confidence:** 3

**Summary:**

This paper introduces KRIS-Bench (Knowledge-based Reasoning in Image-editing Systems Benchmark), a diagnostic tool for evaluating the knowledge-based reasoning capabilities of instruction-based image editing models. While current models can produce visually plausible results, their ability to perform edits that require real-world knowledge remains underexplored.

**Dataset Code Accessibility:**

Yes

**Ethical Considerations:**

No, there are no or only very minor ethics concerns

**Final Justification:**

Thank you to the authors for the rebuttal and discussion. Most of my concerns have been addressed, and I will be increasing my rating accordingly. Additionally, I strongly encourage the authors to update their final draft based on the rebuttal.

**Limitations Weaknesses:**

- The benchmark uses GPT-4o as the automated model to score the generated results. This introduces a potential conflict of interest, as GPT-4o is also evaluated as a top-performing model within the benchmark itself. This may lead to a self-enhancement bias, which could affect the fairness of the comparison.
- The comparison features three closed-source models. To provide a more comprehensive and competitive analysis of the current state-of-the-art, the study would be strengthened by including other prominent closed-source image editing models, such as FLUX.1 Kontext, SeedEdit, and so on.
- The paper conducted a user study to validate the reliability of the VLM-based scores by comparing them against human expert ratings. Providing a per-model breakdown of the user study results for each of the four evaluation metrics (Visual Consistency, Visual Quality, Instruction Following, and Knowledge Plausibility) would be helpful.
- The paper mentioned in the "Data Collection" section that the data is mainly collected from the Internet, a small part is generated by the generative model, and ChatGPT is used to enhance the instructions. This mixed-source data collection process may inadvertently introduce bias.

**Strengths Contributions:**

- The paper is well-written and organized. The abstract and introduction clearly define the research gap and the paper's contributions. The figures and tables are highly informative and well-designed.
- The work's most impactful contribution may be its innovative evaluation protocol. It introduces a novel metric, Knowledge Plausibility (KP), which directly assesses whether an edit aligns with real-world scientific, cultural, or commonsense knowledge—a critical but often-overlooked aspect of intelligent editing.
- The paper's taxonomy (factual, conceptual, procedural knowledge) is borrowed from educational theory, which is a novel perspective.

---

> ### Author Rebuttal · Authors · 2025-07-31
>
> We sincerely appreciate you for constructive suggestions to improve the paper. We hope our response can address your concerns:
>
> **1. On the use of GPT-4o as a scoring model and potential bias**
>
> We understand the concern regarding possible bias due to GPT-4o being both an evaluator and a participant in the benchmark. **To address this, we include evaluation based on an open-source vision-language model, Qwen2.5-VL-72B (Supplementary Material Section E).** As shown in Table 3 and Figure 7, the ranking and task-level scoring trends of Qwen2.5-VL-72B closely align with those produced by GPT-4o, reinforcing the consistency of our evaluation protocol.  We will clarify this in the revision for better illustration.
>
> **2. On inclusion of closed-source models such as FLUX.1 Kontext and SeedEdit**
>
> We appreciate the suggestion: including more closed-source models would indeed make our evaluation results more comprehensive. **The FLUX.1 Kontext series models were released after our submission deadline.** Below are the results of the open-source version, FLUX.1 Kontext [dev], and the closed-source version, FLUX.1 Kontext [pro]/[max]. Due to space limitations, we only report the average scores across each knowledge dimension:
>
> | Model     | Factual Knowledge | Conceptual Knowledge | Procedural Knowledge | Overall Score |
> |------------------------|-------------------|-----------------------|-----------------------|---------------|
> | Flux.1 Kontext [dev]   | 53.28             | 50.36                 | 42.53                 | 49.54         |
> | Flux.1 Kontext [pro]   | 58.14             | 55.06                 | 46.69                 | 54.17         |
> | Flux.1 Kontext [max]   | 59.04            | 57.22                 | 45.60                 | 55.12         |
>
> These results demonstrate that Flux.1 Kontext series models exhibit strong reasoning-based image editing capabilities, outperforming all other open-source models yet falling behind close-source models like Doubao, Gemini 2.0, and GPT-4o.
>
> **Regarding SeedEdit, this model is already included in our paper as "Doubao".** We will clarify this in the revision.
>
> We will also continue maintaining and updating the leaderboard as more high-performing models become accessible.
>
> **3. On providing user study breakdown across models and metrics**
>
> Thank you for the suggestion. We have prepared a summary table presenting the user study results for three top performing models across the four evaluation metrics:
>
> | Metric                 | Model   | Pearson r | MAE    |
> | ---------------------- | ------- | --------- | ------ |
> | Visual Consistency     | GPT4o     | 0.4649    | 0.8130 |
> |                        | Gemini 2.0| 0.8053    | 0.4463 |
> |                        | Doubao| 0.9214    | 0.3141 |
> | Visual Quality         | GPT4o| 0.7097    | 0.1915 |
> |                        | Gemini 2.0| 0.6170    | 0.2740 |
> |                        | Doubao| 0.6602    | 0.2821 |
> | Instruction Following  | GPT4o     | 0.6514    | 0.5227 |
> |                        | Gemini 2.0| 0.6065    | 0.6301 |
> |                        | doubao  | 0.7479    | 0.6287 |
> | Knowledge Plausibility | GPT4o     | 0.9467    | 0.6815 |
> |                        | Gemini 2.0| 0.9343    | 0.5559 |
> |                        | Doubao| 0.9587    | 0.4802 |
>
> As shown in the table, the correlation between human ratings and VLM-based scores for Visual Consistency is high for Gemini 2.0 (0.8053) and Doubao (0.9214), but noticeably lower for GPT-4o (0.4649). We attribute this to GPT-4o’s tendency to subtly enhance image details or alter the overall style and color tone, changes that may go unnoticed by VLMs but are often deemed important by human evaluators.
>
> In contrast, for Visual Quality, Instruction Following, and Knowledge Plausibility, all three models exhibit stronger correlations (from 0.6065 to 0.9587) between VLM and human ratings, demonstrating the reliability of VLM-based evaluation on these aspects.
>
> In addition, we present the correlation between the evaluation results from Qwen2.5-VL-72B and the user study as follows:
>  | Metric          | Model  | Pearson r | MAE    |
> | -------------------------- | ------ | --------- | ------ |
> | Visual Consistency     | GPT4o    | 0.5626    | 1.2847 |
> |                            | Gemini 2.0 | 0.7693    | 0.9678 |
> |                            | Doubao | 0.9014    | 0.5865 |
> | Visual Quality         | GPT4o   | 0.6138    | 0.2226 |
> |                            |  Gemini 2.0| 0.6248    | 0.3179 |
> |                            | Doubao | 0.6184    | 0.2877 |
> | Instruction Following  | GPT4o    | 0.5346    | 0.6854 |
> |                            |  Gemini 2.0| 0.6481    | 0.7702 |
> |                            | Doubao | 0.6731    | 0.7862 |
> | Knowledge Plausibility | GPT4o    | 0.8784    | 0.6069 |
> |                            |  Gemini 2.0| 0.9063    | 0.6360 |
> |                            | Doubao | 0.9147    | 0.5854 |
>
> The results show that, in most cases, the scoring by Qwen2.5-VL-72B remains highly correlated with human judgments. Notably, for the Visual Consistency of GPT-4o, Qwen2.5-VL-72B demonstrates a higher correlation (0.5626) with human evaluations than GPT-4o's own scores. This may be attributed to a potential bias of GPT-4o in favor of its own generated outputs, as you rightly pointed out in your review comments. We will further elaborate on this issue in the revised version.
>
> By including evaluation results from two different models and providing a more detailed user study analysis, we believe this will help readers better understand the findings. Thank you again for your insightful suggestion.
>
> **4. On potential bias introduced by mixed data sources**
>
> We appreciate the your concern regarding potential bias introduced by mixing Internet-sourced data, model-generated content, and ChatGPT-augmented instructions. This is indeed a critical issue when constructing benchmarks. Our data construction strategy aligns with recent practices in benchmark development, where generative tools are often leveraged to supplement hard-to-source or domain-specific content.
>
> Importantly, the proportion of generative content in our dataset is minimal, less than 5%, and is limited to a few specific sub-tasks ( anomaly correction and chemistry), where certain types of images (e.g., rare human anomalies) are difficult to obtain reliably from the Internet. We used model-generated data solely to ensure completeness and coverage in these niche domains.
>
> To address the concern about potential bias introduced by mixing AIGC and real images, we will explicitly separate the performance evaluation on these subsets in our revision. This will allow for a clearer understanding of any differences introduced by synthetic data and help avoid conflating their effects with those of real-world data.

---

> > ### Comment · Reviewer_4uTV · 2025-08-03
> >
> > Thank you for the authors’ comprehensive response. I believe my concerns have been adequately addressed. However, I have a few questions I would like to discuss with the authors.
> >
> > As noted in the paper — *"Based on Table 2, nearly all models consistently perform the weakest on procedural knowledge"* — this work indeed highlights a key limitation of current image editing models in understanding the physical world. Inspired by the use of prompt refiner in video generation models, I wonder whether enhancing user prompts with knowledge-driven augmentation using LLMs could help mitigate this issue to some extent.
> >
> > Additionally, after reading the other reviews, I realized I share a similar concern: while the proposed categorization of knowledge is interesting, the authors have not sufficiently explained its advantages or necessity. Firstly, the proposed categorization does not offer a clear, mutually exclusive, and exhaustive framework for task classification. Many tasks exhibit overlapping characteristics — for example, "making someone wear a specific brand of clothing" involves both factual knowledge and procedural knowledge. Secondly, the proposed categorization implicitly assumes that each task has a single correct answer. However, many image editing tasks are open-ended in nature — for example, "make this food look more delicious" — and such tasks are not well accounted for in the current framework. Finally, the authors claim that *"Compared to previous benchmarks, our knowledge-based taxonomy better captures how humans intuitively represent, decompose, and solve editing tasks."* However, this assumes that humans approach editing tasks in a cognitively hierarchical manner. In practice, humans often classify tasks based on their goals, and task decomposition tends to be goal-driven rather than strictly aligned with cognitive categories.

---

> > > ### Author Response · Authors · 2025-08-04
> > > **Response to Reviewer 4uTV (2/3)**
> > >
> > > **2. Advantages and Necessity of Our Framework**
> > >
> > > We have thoroughly discussed the advantages and necessity of our framework in our response to Reviewer MFZ4 (see "1. Clarifying Our Motivation and Comparison with RISEBench (1)").
> > >
> > >  In addition, we address your specific concerns as follows:
> > >
> > > **(1) Framework categorization and Overlap Issue**
> > >
> > > We believe this concern arises from a misunderstanding. Our categorization is based on the primary cognitive demand of each task. Procedural tasks inherently involve multi-step planning and reasoning, and each step may indeed involve different types of knowledge (e.g., factual or conceptual), this does not imply overlap at the classification level. Our classification hinges on the overall cognitive load required by the task, not the presence of incidental knowledge types in sub-steps.
> > >
> > > As you mentioned in your examples, “making someone wear a specific brand of clothing” is fundamentally a procedural task. While it may involve factual knowledge at certain steps, for instance, knowing which body part the clothing should go on, or recognizing what a particular brand’s clothing looks like, the overall reasoning process that integrates these steps is primarily procedural in nature. Therefore, it is clearly and justifiably categorized under Procedural Knowledge.
> > >
> > > Another, even more intuitive example is the Multi-Instruction Execution scenario defined in our paper. For example, generating a poster from an image of a hamburger often involves a prompt with multiple, complex steps. Each individual step may draw on more basic categories of knowledge, but the overall generation process remains fundamentally procedural.
> > >
> > > Additionally, as elaborated in our response to Reviewer MFZ4, we provided a detailed comparison between our framework and the contemporaneous work RISEBench, which demonstrates the orthogonal and unambiguous task categorization in our framework (see "1. Clarifying Our Motivation and Comparison with RISEBench (2)").
> > >
> > > **(2) Involve Open-Ended Tasks**
> > >
> > > KRIS-Bench does not enforce a single correct answer. Instead, we adopt VLM-based scoring, which is inherently capable of accommodating diverse and valid outputs. This design choice ensures flexibility while maintaining evaluation rigor.
> > >
> > > While some tasks may appear more open-ended, such as making food “more delicious,” there is still an underlying layer of common-sense or cultural knowledge involved. People typically associate specific attributes (for example, golden-brown texture or visible steam) with better taste. In this sense, such tasks remain compatible with our knowledge-centric framework. However, “deliciousness” is highly subjective and culturally variable, which makes it less suitable as a diagnostic benchmark item.
> > >
> > > We have intentionally selected tasks with clearly defined goals and cognitively interpretable, in order to ensure clarity in what needs to be evaluated. Many of our editing instructions are relatively specific, such as “move the food from the left to the center of the table” or "add a piece of solid sodium to the water". This specificity supports a more targeted assessment. It is not a limitation of our framework, but rather a design choice to make evaluation concrete and reproducible.
> > >
> > > Importantly, our use of a vision-language model scoring naturally supports output diversity. A well-constructed prompt that admits multiple plausible outputs can still be handled robustly within our evaluation setup. While our current tasks emphasize more objective editing goals, the framework is fully capable of accommodating open-ended instructions, provided they involve interpretable reasoning and knowledge application.

---

> > > ### Author Response · Authors · 2025-08-04
> > > **Response to Reviewer 4uTV (3/3)**
> > >
> > > **(3) Cognitive Levels vs. Goal-Driven Categorization**
> > >
> > > We appreciate the your insightful concern. We fully agree that humans often approach tasks with goal-driven intentions. However, we believe that goal-driven planning and cognitively structured execution are not mutually exclusive. In fact, achieving a given goal necessarily requires the deployment of specific cognitive processes, regardless of how the goal is initially formulated. Goal-driven categorization implicitly tests cognitive abilities, even if not explicitly framed as such. For example, an atomic editing task such as "changing the color of an object" might appear purely goal-driven, yet under our framework, it reflects the application of factual knowledge, such as recognizing the object and understanding its visual attributes.
> > >
> > > However, our framework does not dispute that goals drive behavior. Rather, it aims to provide a systematic and interpretable lens for analyzing the cognitive operations that models must perform to achieve those goals. By categorizing tasks based on their dominant cognitive requirements, we gain a principled way to evaluate and diagnose whether models possess the underlying reasoning skills required for generalization across goal types.
> > >
> > > Moreover, even within goal-driven tasks, the required reasoning can vary significantly in difficulty. Inspired by your comment, we reflected further on this issue and conducted a comparative experiment. Take the atomic operation "add" as an example. The instruction might be phrased as "add a panda" or "add China’s national treasure", both referring to the same species. While most models succeed in the first case, some fail in the second. Under a purely goal-driven categorization, such discrepancies are ambiguous and difficult to interpret. It remains unclear why certain add operations succeed while others do not. This phenomenon is also illustrated in our examples in Figure 3 (c) and (d). While most models are capable of performing surface-level edits such as changing animals or altering colors, the resulting outputs are often incorrect or implausible. This indicates that their knowledge-based reasoning abilities remain limited.
> > >
> > > In contrast, our taxonomy provides a cognitive-level explanation for this phenomenon. The model fails not because of the goal itself, but because it lacks conceptual knowledge, specifically the ability to associate abstract or symbolic references like "China's national treasure" with their intended visual entities, such as a panda. In this sense, our taxonomy complements goal-driven perspectives by adding a cognitive dimension to task understanding, enabling more precise analysis of how a model succeeds or fails, not just what it tries to do. We will elaborate on this point more clearly in the revised version. Thank you again for your comments.
> > >
> > >
> > >
> > > We hope our response helps clarify your concerns. And we are also happy to address any questions you may have after reading our response.

---

> > > > ### Comment · Reviewer_4uTV · 2025-08-05
> > > >
> > > > Thank you to the authors for the rebuttal and discussion. Most of my concerns have been addressed, and I will be increasing my rating accordingly. Additionally, I strongly encourage the authors to update their final draft based on the rebuttal.

---

> > > > > ### Author Response · Authors · 2025-08-05
> > > > >
> > > > > Thank you very much for your insightful comments. We're glad to hear that most of your concerns have been addressed, and we sincerely appreciate your updated rating. We will make sure to incorporate your suggestions and the points raised in the rebuttal into the final version of the paper. Your feedback has been invaluable in shaping the revision.

---

> > ### Author Response · Authors · 2025-08-04
> > **Response to Reviewer 4uTV (1/3)**
> >
> > Thank you for your response, we’re glad to hear your earlier concerns have been addressed. Below is our reply to your current questions:
> >
> > **1.  Enhancing user prompts with knowledge-driven augmentation using LLMs**
> >
> > Thank you for your insightful suggestion. We fully agree that enhancing user prompts with knowledge-driven augmentation using LLMs could indeed help mitigate the performance gap in conceptual/procedural knowledge tasks.
> >
> > To prove this, we evaluated the generative model Bagel [1], which is equipped with textual reasoning capabilities. We found that incorporating an explicit “think” step during image generation can significantly enhance model performance on knowledge-intensive image editing tasks. This process is akin to a prompt refiner. For example, below are two outputs from Bagel on tasks involving biology and chemistry:
> >
> > - The user wants to demonstrate the effect of planting hydrangeas in acidic soil on their color. **The input image shows hydrangeas with mixed blue and purple hues, which are influenced by soil pH. The output image should depict a clear color shift—hydrangeas turning predominantly pink or red—reflecting the impact of acidic conditions.** The structure and layout should remain consistent, with only the flower color changed.
> >
> > - The task involves adding barium chloride to a sulfuric acid solution. **This reaction produces barium sulfate, a white precipitate. The answer image should visually show the addition of barium chloride and the formation of the white precipitate.** The original bottle and liquid should remain unchanged, except for the visible appearance of the precipitate in the solution.
> >
> > After introducing the textual reasoning process, we observed that the original prompts were rewritten in a way that enabled the generation of correctly edited images.
> >
> > We present Bagel’s score in KRIS-Bench here:
> >
> > | Model | Factual Knowledge | Conceptual Knowledge| Procedural Knowledge | Overall Score |
> > |----|----|----|------|---|
> > | Bagel | 60.26 | 55.86 | 51.69 | 56.21 |
> > | Bagel with think | 66.18 | 61.92 | 49.02 | 60.18 |
> >
> > We can clearly see that explicitly incorporating the reasoning process to reformulate prompts led to significant improvements across all knowledge dimensions.
> >
> > Another image editing model, HiDream-E1 [2], adopts an explicit prompt refinement process. Before editing the image, it leverages a vision-language model (VLM) to rewrite the input prompt into a refined caption, which serves as the basis for the subsequent editing (see the official GitHub implementation). The evaluation results are as follows:
> >
> > | Model | Factual Knowledge | Conceptual Knowledge| Procedural Knowledge | Overall Score |
> > |------|----|--------|---------|-----|
> > | HiDream-E1 | 43.31 | 50.05 | 37.64 | 44.72 |
> >
> > The results show that the model achieves strong performance in the Conceptual Knowledge dimension, which demands substantial prior knowledge. It surpasses all open-source baselines reported in the original paper, highlighting the effectiveness of the prompt refinement strategy.
> >
> > We will include these models' results and provide a more detailed discussion with a potential solution to solve reasoning editing tasks. Thank you for your insightful question!
> >
> > [1] Emerging Properties in Unified Multimodal Pretraining
> >
> > [2] HiDream-I1: A High-Efficient Image Generative Foundation Model with Sparse Diffusion Transformer

---

### Official Review · Reviewer_MFZ4 · 2025-07-06

**Ethics Flags:** Discrimination, bias, and fairness
**Rating:** 5
**Confidence:** 3

**Summary:**

The motivation of the manuscript is not sufficiently articulated. Although it proposes a task categorization based on knowledge types, it does not adequately explain the advantages and necessity of this new categorization. In addition, the manuscript lacks a clear justification for the selection of 22 tasks and how these tasks sufficiently cover practical editing scenarios. The visualization examples mainly focus on objects and animals, with limited coverage of human-centric or face-related editing tasks, which may reduce the relevance to certain real-world applications.

**Additional Feedback:**

1.The current manuscript proposes a task taxonomy based on factual, conceptual, and procedural knowledge, inspired by educational theory. However, it lacks a comparative argument demonstrating why this framework is superior to the temporal, causal, spatial and logical reasoning attributes adopted in RISEBench. It is recommended that the authors systematically elaborate in the Introduction on how these three knowledge types align with human cognitive reasoning processes and the requirements of image editing tasks.
2.The notion of "fine-grained reasoning" should be clearly defined, specifying in what aspects the proposed framework provides finer granularity (e.g., task granularity, knowledge types, reasoning hierarchy).
3. The authors defined 22 editing tasks; however, the manuscript does not provide a clear explanation of the rationale for choosing this specific number or the criteria used for task selection. It is recommended to provide a clearer justification for the number of tasks and its theoretical basis, including an explanation of why 22 tasks were chosen over a larger or smaller set.
4. Compared to the diverse and complex image editing scenarios in the real world, the authors are encouraged to discuss whether the editing tasks in KRIS-Bench sufficiently cover practical needs. They should also consider extending the experiments to further validate the model’s generalization capability.
5. Figure 3 mainly presents editing examples of objects and animals, offering relatively limited coverage. It is recommended to include visual examples of featuring human figures or face editing, such as expression changes, pose adjustments, or identity transformations.
6. The current task addresses a specific cultural context, exemplified by the “national treasure panda,” while future work should consider expanding to multilingual and multicultural task sets.

**Dataset Code Accessibility:**

Yes

**Ethical Comments:**

The author has addressed some ethical issues, such as the paper's statement that the data mainly comes from publicly available images on the Internet, existing datasets, and a small portion is synthesized by generative models. However, there are potential concerns about cultural biases in task design and the social impact of promoting knowledge-based image editing models, especially regarding the risk of abuse.

**Ethical Considerations:**

No, there are no or only very minor ethics concerns

**Final Justification:**

Thank you for your detailed response. Most of my concerns have been addressed, and I will increase the score accordingly.

**Limitations Weaknesses:**

The motivation of the manuscript is not sufficiently articulated. Although it proposes a task categorization based on knowledge types, it does not adequately explain the advantages and necessity of this new categorization. In addition, the manuscript lacks a clear justification for the selection of 22 tasks and how these tasks sufficiently cover practical editing scenarios. The visualization examples mainly focus on objects and animals, with limited coverage of human-centric or face-related editing tasks, which may reduce the relevance to certain real-world applications.

**Strengths Contributions:**

The manuscript presents a novel and well-motivated benchmark, KRIS-Bench, for evaluating the reasoning capabilities of instruction-based image editing models. By drawing on educational cognitive theory, it introduces a structured taxonomy of factual, conceptual, and procedural knowledge types, offering a fresh perspective beyond prior task. The benchmark covers 22 tasks across 7 reasoning dimensions, and the introduction of the Knowledge Plausibility metric represents an important step towards assessing whether model outputs align with real-world knowledge. The extensive experiments on nine state-of-the-art models are thorough and reveal meaningful insights into current limitations in knowledge-grounded reasoning for image editing.

---

> ### Author Rebuttal · Authors · 2025-07-31
>
> Thank you for the helpful feedback. We're glad that you found the motivation of KRIS-Bench clear, the taxonomy of factual, conceptual, and procedural knowledge well-structured, and the Knowledge Plausibility metric insightful. We also appreciate your recognition of our experimental design and presentation.
>
> We hope our response can address your concerns:
>
> **1. Clarifying Our Motivation and Comparison with RISEBench**
>
>   **(1) The motivation and necessity of our taxonomy**
>
> Our taxonomy is grounded in established cognitive educational theories, the revised Bloom’s taxonomy proposed by Anderson and Krathwohl (2001). From a cognitive alignment perspective, this framework organizes knowledge hierarchically to facilitate increasingly complex mental operations. Applied to instruction-based image editing, these knowledge types naturally map onto distinct reasoning requirements, which models must satisfy to effectively interpret and execute diverse user-instruction edits over corresponding images.
>
> Factual knowledge aligns with fundamental cognitive skills of remembering and basic understanding, involving the recognition and recall of specific observable details. In the context of image editing, this manifests clearly in tasks requiring precise manipulation of visual elements, such as changing object colors or repositioning objects based on explicit spatial cues. Such factual tasks constitute the foundational cognitive building blocks without which higher-level reasoning cannot occur reliably.
>
> Conceptual knowledge corresponds to higher-order cognitive skills, including deeper understanding, generalization, and application. At this cognitive stage, humans grasp abstract relationships, discern underlying patterns, and apply principles across varying contexts. In image editing, conceptual tasks require models to accurately interpret and implement modifications reflecting nuanced socio-cultural contexts or scientific principles, such as inducing chemical transformations. Such tasks demand an abstract understanding of underlying concepts, facilitating generalization across diverse editing scenarios.
>
> Procedural knowledge reflects advanced cognitive abilities involving analysis, evaluation, and creative synthesis. At this stage, humans cognitively break down complex scenarios, evaluate among competing solutions, and synthesize novel, contextually coherent outcomes. In the realm of image editing, procedural tasks necessitate multi-step reasoning, structured task decomposition, and execution based on intricate logical rules. Representative examples include complex logical puzzles, coordinated multi-element image manipulations, or generating edits that demand detailed planning and adaptive reasoning strategies.
>
> Compared to previous benchmarks, our knowledge-based taxonomy better captures how humans intuitively represent, decompose, and solve editing tasks. Therefore, our framework is necessary for constructing a benchmark that enables human-aligned evaluations for multimodal generative systems.
>
> **(2) The advantages of our framework to RISEBench**
>
> Compared to RISEBench, KRIS-Bench offers three key advantages:
>
> **Explicit Cognitive Grounding.** KRIS-Bench leverages the widely recognized cognitive model from Anderson and Krathwohl’s revised Bloom’s taxonomy. Unlike attribute-based taxonomies relying on external task characteristics such as spatial or causal interactions, our taxonomy explicitly corresponds to fundamental cognitive processes. This clear cognitive grounding ensures each task category uniquely reflects distinct human reasoning operations, thereby enhancing both the interpretability of evaluation outcomes and alignment with human cognitive reasoning patterns.
>
> **Orthogonal and Unambiguous Task Categorization.** The hierarchical knowledge structure naturally defines clear, orthogonal task boundaries, reducing ambiguity and overlap. By contrast, RISEBench’s attribute-based taxonomy frequently suffers from unclear boundaries and overlaps between categories. For example, tasks labeled "causal reasoning" often implicitly involve temporal or spatial reasoning, making it challenging to precisely assign tasks to single categories or interpret the cognitive demands of each task.
>
> **Enhanced Evaluation Metrics.** The hierarchical cognitive structure facilitates systematic expansion and fine-grained differentiation of tasks. New editing scenarios can be seamlessly integrated by explicitly defining their cognitive prerequisites according to established knowledge types. This structured scalability also supports our Knowledge Plausibility (KP) metric, which evaluates models’ adherence to factual accuracy and conceptual validity. KP can robustly differentiate whether edits merely follow superficial instructions or demonstrate genuine understanding of domain-specific knowledge. As new tasks are introduced, the metric remains consistently applicable, ensuring coherent and robust evaluation across evolving benchmarks.
>
> In summary, our taxonomy's explicit cognitive grounding, orthogonal categorization, and robust scalability provide a more theoretically coherent, precise, and practically effective benchmark framework compared with attribute-based approaches such as those used in RISEBench.
>
> **2. Definition of Fine-Grained Reasoning**
>
> In KRIS-Bench, "fine-grained" refers to our benchmark's ability to evaluate reasoning capabilities at multiple, detailed levels rather than at a single task level. Specifically, our framework achieves finer granularity across three dimensions: knowledge types, by explicitly distinguishing among Factual, Conceptual, and Procedural knowledge; reasoning dimensions, by systematically defining seven distinct categories; and task granularity, by further decomposing these reasoning dimensions into 22 carefully designed editing tasks, each precisely targeting a specific cognitive skill required by image editing models. Consequently, this fine-grained structure enables our evaluation protocol to precisely measure model performance across different knowledge types and cognitive demands.
>
> **3. Justification of 22 Tasks Selection**
>
> Based on our theoretical framework, the selection of 22 tasks was finalized through a comprehensive analysis of existing image-editing benchmarks and practical application scenarios. Specifically, we adhered to the following criteria when selecting tasks:
>
> **(1) Task Diversity**
>
> We ensured that each knowledge type includes multiple representative tasks. These tasks cover different cognitive difficulty levels, from basic factual tasks through more complex conceptual tasks, to procedural tasks requiring fine-grained planning and reasoning. Such diverse coverage provides a comprehensive evaluation of model capabilities across multiple cognitive dimensions.
>
> **(2)  Efficiency Tradeoff**
>
>  We carefully considered practical efficiency during benchmark evaluation. A smaller number of tasks would be insufficient to capture the diversity of model capabilities, while too many tasks might introduce redundancy, increasing evaluation complexity without significantly enhancing diagnostic value. Through iterative testing and analysis, we identified 22 tasks as the optimal scale, effectively capturing diverse cognitive skills combinations while avoiding redundancy.
>
> **(3) Task Extensibility**
>
> Our hierarchical taxonomy, grounded in Bloom’s cognitive framework, provides a well-structured foundation for future task expansion. The current 22 tasks do not represent a fixed limitation of our framework; rather, they constitute an initial set suitable for the current stage. As models advance and application scenarios evolve, our framework readily accommodates additional tasks.
>
> Considering these aspects collectively, we established 22 tasks as the optimal initial set, effectively balancing evaluation comprehensiveness, task diversity, operational efficiency, and future extensibility.
>
>
> **4&5. Coverage of Real-World Cases and Lack of Human-Centric Examples**
>
> KRIS-Bench indeed incorporates tasks closely related to practical, real-world editing scenarios, particularly addressing human-centric editing. For example, under Factual Knowledge, the "Anomaly Correction"  taskscover scenarios frequently encountered in practical image editing, such as correcting abnormalities (e.g., unnatural postures). Similarly, in the Conceptual Knowledge, our "Medicine" task directly engages realistic human-body editing scenarios, such as visually simulating healing processes after fractures, clearly linking our tasks to medical visualization applications. Additionally, under Procedural Knowledge, the "Multi-element Composition" task explicitly targets human-centric applications, such as altering clothing or combining fashion items on human figures, a scenario common in virtual try-on and personalized advertising.
>
> We fully acknowledge the your observation that our current visualization examples focus predominantly on objects and animals, offering relatively limited visual coverage of these human-centric or face-related editing tasks. In our revision, we will expand Figure 3 by including additional visual examples involving human figures. This will better highlight KRIS-Bench’s existing relevance to human-related real-world applications and further validate the models’ generalization capability in diverse, practical editing scenarios.
>
> **6. Cultural and Multilingual Considerations**
>
> Currently, we have made efforts to include examples from different countries, religions, and cultural backgrounds under the Humanities category. We fully agree that incorporating more diverse linguistic and cultural contexts would significantly enhance KRIS-Bench's generalizability and international relevance. We have also added a discussion in the Limitations section and plan to further expand the task pool in future versions to better reflect diverse cultural contexts.

---

### Note · Authors · 2025-08-15

We thank all five reviewers for their constructive feedback and for recognizing KRIS-Bench’s novelty, cognitive grounding, and evaluation design. Across the initial reviews, concerns included clarifying the motivation and necessity of our knowledge-based taxonomy (`MFZ4`, `4uTV`), justifying the selection of 22 tasks and their coverage of real-world scenarios, especially human-centric ones (`MFZ4`), addressing potential overlaps and handling of open-ended tasks (`4uTV`), mitigating possible bias from using GPT-4o as both participant and evaluator (`4uTV`), providing finer-grained human–VLM correlation analysis (`hCAc`), and improving figure readability and dataset scalability (`MFZ4`, `tbw8`).

In our rebuttal, we elaborated on the cognitive foundations of our taxonomy and its advantages over attribute-based frameworks such as RISEBench, outlined our task selection process as a balance of diversity, efficiency, and extensibility, and reaffirmed the inclusion of human-centric tasks with plans to expand visual examples. In response to reviewers’ comments, we incorporated Bagel’s “think” process and HiDream-E1’s prompt-refinement approach, achieving measurable performance gains. We also verified scoring robustness using the open-source Qwen2.5-VL-72B, and provided per-model and per-metric human–VLM correlation tables alongside analyses of key failure modes. To make our analysis more comprehensive, we reported results of FLUX.1 Kontext series models。

During the discussion phase, reviewers (`4uTV`, `tbw8`, `hCAc`, `68Wp`) acknowledged that major concerns had been addressed. Reviewers `4uTV` and `hCAc` raised their ratings, Reviewer `tbw8` thinks “this paper will be valuable to the community and recommend acceptance”, and Reviewer `68Wp` maintained a positive score of accepting. We appreciate the recognition of our clearer motivation, strengthened evaluation protocol, and expanded experiments, as well as support for future cultural and multilingual extensions.

 We will incorporate all clarifications, new results, and figure and layout improvements into the camera-ready version, further establishing KRIS-Bench as a principled, extensible, and practically relevant benchmark for advancing knowledge-based reasoning in instruction-based image editing.

---

### Decision · Program_Chairs · 2025-09-18

**Decision:**

Accept (poster)

**Comment:**

This is a nice contribution to image editing benchmarking particularly because it offers a different set of cognitive criteria to break down model performance, which can be valuable for understanding model performance and how to improve it to align better with humans. That being said, reviewers still felt there are some limitations including lack of multi-turn editing and potential biases. I think the reviewers were all positive about the work though after the rebuttal, and I agree with their assessment.